



# Evolution of layered density and microstructure in near-surface firn around Dome Fuji, Antarctica

Ryo Inoue[1], Shuji Fujita[1,2], Kenji Kawamura[1,2,3], Ikumi Oyabu[2], Fumio Nakazawa[1,2], Hideaki Motoyama[1,2]

[1]The Graduate University for Advanced Studies, SOKENDAI, Tokyo 190–8518, Japan
[2]National Institute of Polar Research, Tokyo 190–8518, Japan
[3]Japan Agency for Marine–Earth Science and Technology, Kanagawa 237–0061, Japan

*Correspondence to*: Ryo Inoue (inoue.ryo@nipr.ac.jp)

**Abstract.**

To better understand the near-surface evolution of polar firn in low accumulation areas (<30 mm w.e. yr$^{-1}$), we investigated the physical properties: density, microstructural anisotropy of ice matrix and pore space, and specific surface area (SSA), of six firn cores collected within 60 km around Dome Fuji, East Antarctica. The physical properties were measured at the intervals of ≤0.02 m over the top 10 m of the cores. The main findings are: (i) lack of significant density increase in the top ~4 m, (ii) lower density near the dome summit (~330 kg m$^{-3}$) than the surrounding slope area (~355 kg m$^{-3}$), (iii) developments of vertically elongated microstructure and its contrast between layers within the top ~3 m, (iv) more pronounced vertical elongation at sites and periods with lower accumulation rates than those with higher accumulation rates, (v) rapid decrease in SSA in the top ~3 m, and (vi) lower SSA at lower accumulation sites, but this trend is less pronounced than that of microstructural anisotropy. These observations can be explained by the combination of the initial physical properties on the surface set by wind conditions and the metamorphism driven by water vapor transport through the firn column under a strong vertical temperature gradient (temperature gradient metamorphism, TGM). The magnitude of TGM depends on the duration of firn layers under temperature gradient, determined by accumulation rate; longer exposure causes a more vertically elongated microstructure and lower SSA. Overall, we highlight the significant spatial variability in the near-surface physical properties over the scale of ~100 km around Dome Fuji. These findings will help better understand the densification over the whole firn column and the gas trapping process in deep firn, and possible difference in them between existing deep ice cores and the upcoming "Oldest-Ice" cores collected tens of kilometers apart.

## 1 Introduction

Understanding the physical characteristics of near-surface firn is important to many glaciological studies. For example, accurate density profile of near-surface firn is essential to derive the surface mass balance from the change in surface height



observed by satellite altimetry (e.g., Zwally et al., 2015; Alexander et al., 2019). The physical properties are also linked to the radiative characteristics in firn; for example, grain size affects surface albedo (Wiscombe and Warren, 1980), microwave emission, and microwave reflection (Picard et al., 2014), and the number of layers per unit depth is linked to microwave polarization ratio (Surdyk and Fily, 1995). Furthermore, the density evolution over the entire firn column is affected by the

density, microstructure (the shapes and arrangements of ice matrix and pore space), and their layering in the near-surface firn (Alley et al., 1982; Gerland et al., 1999; Freitag et al., 2004; Fujita et al., 2009, 2014, 2016), along with temperature, overburden pressure, and impurity concentrations (e.g., Herron and Langway, 1980; Hörhold et al., 2012; Fujita et al., 2016). The resulting mean density and layering in deep firn determine the depth range of air enclosure into bubbles (e.g., Schwander, 1989; Mitchell et al., 2015).

Environmental conditions in the atmosphere and at the ice sheet surfaces determine the initial physical properties of firn. The deposition of snow or frost may form a firn layer, which is sometimes altered by wind-driven snow redistribution (e.g., Kameda et al., 2008) and fragmentation (e.g., Domine et al., 2009). The layers in the firn column generally exhibit seasonal cycles in physical properties (e.g., higher density in summer than in winter) in areas with high accumulation rate ($> \sim 50$ mm w.e. yr$^{-1}$,

Laepple et al., 2016), suggesting continuous accumulation throughout the year. In contrast, in low-accumulation areas ($< \sim 50$ mm w.e. yr$^{-1}$, represented by dome areas on the East Antarctic Plateau), the layers are tens of centimeters thick and do not show seasonal cycles (Hörhold et al., 2012). This suggests that snow redistribution and precipitation intermittency disrupt the seasonal deposition of layers in low-accumulation areas. After the deposition, the firn layers undergo metamorphism over time by packing and rounding of snow grains, whose rates primarily depend on the firn temperature (e.g., Colbeck, 1989). Moreover,

varying surface heating by diurnal and seasonal variations in insolation produce vertical temperature gradient (TG) in the top few meters, facilitating metamorphism through efficient vertical water vapor transport, where vapor sublimates from warmer grains and condenses to colder grains (e.g., Yosida, 1955; Colbeck, 1983; Pinzer et al., 2012). This process is known as temperature gradient metamorphism (TGM). Accumulation rate is an important factor for the magnitude of TGM because it determines the duration of exposure of a layer to TG (e.g., Hutterli et al., 2009). Wind ventilation may also affect the amount

of sublimation and condensation within the firn by disrupting the saturation levels between the ice matrix and surrounding air (Albert, 2002). Because the above environmental conditions vary over the ice sheets, near-surface firn density and microstructure are also expected to vary. Understanding such variabilities in the physical properties is necessary to better understand differences in firn densification rate and the depth of air enclosure into bubbles at different sites, such as sites for existing deep ice cores and the upcoming "Oldest-Ice" cores tens of kilometers apart (e.g., Fischer et al., 2013; Parrenin et al.,

2017; Obase et al., 2022).

Numerous studies have investigated the physical properties of near-surface firn and their relationship with environmental factors. Near-surface firn densities have been widely measured in Dronning Mord Land, East Antarctica, showing strong dependence of the average density on annual mean wind speed; lower wind speed leads to lower density (e.g., Sugiyama et al.,



2012). Firn cores from multiple polar sites showed that the density variability in the top few meters increases with decreasing mean annual temperature and accumulation rate (within the ranges of 25–180 mm w.e. yr$^{-1}$ and −53–−18 °C, respectively) (Hörhold et al., 2011). Density typically increases with depth due to grain packing and sintering (e.g., Kojima, 1971; Craven and Allison, 1998; Salamatin et al., 2009; Hörhold et al., 2011). However, a few studies reported a lack of significant density increase in the top 2 or 3 m in the interior plateau, e.g., at Point Barnola (Calonne et al., 2017) and Dronning Maud Land plateau (Endo and Fujiwara, 1973; Weinhart et al., 2020), possibly because of small overburden pressure and high viscosity of firn due to low temperature (Endo and Fujiwara, 1973; Calonne et al., 2017) or strong TGM (Alley, 1987).

Typical methods to investigate the microstructure of polar firn are visual inspection of pit walls or image analysis of thin sections of firn samples (e.g., Koerner, 1971; Rick and Albert, 2004; Courville et al., 2007). Recent studies also applied X-ray computed tomography (CT), a high-resolution 3-D imaging technique, to investigate polar firn (Freitag et al., 2004; Fujita et al., 2009; Hörhold et al., 2009; Lomonaco et al., 2011; Linow et al., 2012; Calonne et al., 2017; Moser et al., 2020). The X-ray CT has revealed that polar firn has a vertically elongated structure comprising ice matrix and pore spaces, probably formed by TGM. The previous studies have also reported increase in snow grain size and decrease in snow specific surface area (SSA: the area of ice–pore interface per unit mass of firn) with depth, particularly in the top few meters (e.g., Linow et al., 2012; Calonne et al., 2017; Moser et al., 2020). Some studies also investigated the spatial variability of the near-surface microstructures. Courville et al. (2007) found larger faceted grains in the top 2 m at a site with accumulation hiatus than those at a site with accumulation (<40 mm w.e. yr$^{-1}$) in the Megadune region, East Antarctica. For West Antarctic sites with accumulation rate of 160–200 mm w.e. yr$^{-1}$, lower accumulation rates are associated with more vertically elongated structure in a 15 m deep firn core (Hörhold et al., 2009). These studies suggested that accumulation rate influences the magnitude of TGM by controlling the exposure time of layers under TG in the top few meters. Linow et al. (2012) also reported the variability in SSA profiles at six sites with different accumulation rate (25–180 mm w.e. yr$^{-1}$) and temperatures (−53–−18 °C). Their SSA profiles were reproduced by an SSA evolution model incorporating accumulation rate and snow temperature as inputs, suggesting that near-surface SSA depends on accumulation rate and temperature. In addition, pit wall observations along a traverse route from the coast to Dome C revealed a predominance of rounded grains at the surface of Dome C, and the SSA at Dome C was higher than expected from an empirical SSA–density relationship for seasonal snow (Gallet et al., 2011). Thus, this study concluded that wind increases SSA through snow grain transport, fragmentation, and sublimation (Domine et al., 2009) at the surface of Dome C.

However, most of the previous studies have been conducted in high-accumulation sites (>50 mm w.e. yr$^{-1}$) and key aspects for the evolution of near-surface density and microstructure at low-accumulation sites, such as Dome Fuji, Dome C, or Vostok, remain poorly documented or understood. For example, (i) the inland sites tend to lack density increase in the top few meters, but the mechanisms are uncertain. (ii) Detailed observations of firn microstructure in the top few meters are limited, with only Courville et al. (2007) and Calonne et al. (2017) providing data beyond visual inspection. (iii) The differences in the



developments of density and microstructure of different layers are not well understood. (iv) It is uncertain whether the effects

of accumulation rate, temperature, and wind speed on density and microstructure, observed over the fairly different

environments in previous studies, are noticeable even within smaller environmental changes realistic for low-accumulation

areas (e.g., accumulation rates of 13–35 mm w.e. yr$^{-1}$ during glacial–interglacial cycles at Dome Fuji (Parrenin et al., 2016)).

To address these issues, the density and microstructure must be investigated at high resolution for multiple sites in low-

accumulation areas. However, the near-surface firn in these areas is often too fragile to perform detailed on-site or laboratory

measurements of physical properties. In particular, thin section analysis and X-ray CT measurement typically require thin or

small cylindrical samples, which are technically challenging and time-consuming with the fragile samples. These difficulties

may have hindered continuous or multi-site measurements of the firn properties. Here, several optical or electrical methods

can provide continuous, high-resolution, non-destructive, and time-efficient measurements. For example, gamma-ray

transmission measurements provide high-resolution density data on millimeter scales. Near-infrared (NIR) reflectivity

measurements determine the SSA of snow and firn (e.g., Matzl and Schneebeli, 2006; Arnaud et al., 2011). Tensorial values

of relative permittivity in microwave and millimeter wave frequencies, which can be measured using an open resonator,

provide the proxy for microstructural anisotropy of firn (e.g., Fujita et al., 2009, 2014, 2016). Fujita et al. (2009, 2016) applied

this method to three firn cores drilled near the Dome Fuji Station, but the measurements were limited to depths below 10 m

due to poor core recovery at shallower depths.

In this study, we measured the density, microstructural anisotropy, and SSA of six firn cores drilled at five sites around Dome

Fuji, continuously at high resolution (0.0025–0.02 m) using optical and electrical methods. Our data provide a first detailed

view of the evolutions of layered density and microstructure in the top few meters around Dome Fuji. Moreover, they allow

the comparison of the properties among the five sites with different environments. Based on the new data, we discuss the key

processes for the evolutions in density and microstructure and possible causes for their spatial and temporal variabilities.

## 2 Methods

### 2.1 Study area and samples

In Antarctica, the accumulation rate varies depending on the topography and atmospheric circulation (which influences

moisture transport) from the coast to inland areas. In the Dronning Maud Land, anticyclonic activities play a major role in

transporting heat and moisture from lower latitudes toward Dome Fuji (Fig. 1a). These activities tend to originate from the

eastern Atlantic Ocean and Indian Ocean (e.g., Suzuki et al., 2008; Hirasawa et al., 2013). At the Dome Fuji Station, strong

winds mostly blow from the northeast ($53 \pm 48°$ for wind speeds above 8 m s$^{-1}$, black arrow in Fig. 1b). Since air masses tend

to release their moisture when they blow in the upslope direction due to orographic lift, snow accumulation is higher on the



north eastern side (windward side) of topographical ridge along the northwest direction than on the leeward side (Fujita et al., 2011; Van Liefferinge et al., 2021; Oyabu et al., 2023).

We analyzed six firn cores from five sites around Dome Fuji: NDF, NDFN, DFSE, DFS, and DFNW (Fig. 1b). The information

of the sites and cores is listed in Table 1 and briefly described below. The NDF and NDFN sites are located 54 and 48 km south of the Dome Fuji Station, respectively. Two cores were collected from the NDF site in December 2012 and 2017, located 0.2 km apart and named NDF13 and NDF18, respectively (Oyabu et al., 2023). A core was collected from the NDFN site in December 2018. The DFSE and DFNW sites are located 44 km southeast and 28 km north of the Dome Fuji Station, respectively. A core was collected from each site in 2017–2018. We carefully drilled, handled, and transported the cores to

preserve the physical properties in the top several meters. The good sample conditions allowed us to perform continuous physical measurements on the near-surface firn around Dome Fuji. In addition, we used published data from the DFS10 core collected 9 km south of the Dome Fuji Station (Fujita et al., 2016).

Among our sample locations, the DFNW site has the highest accumulation rate (29.4 mm w.e. yr$^{-1}$ for 1885–1992 C.E.), which

is 24% higher than the lowest accumulation rate at NDF13 (23.7 mm w.e. yr$^{-1}$) (Oyabu et al., 2023). The accumulation rate is the most variable environmental factor among the core sites; other factors, such as 10 m snow temperature, 10 m wind speed, and summer insolation derived from observations or the ERA5 reanalysis, show minor relative differences of 3%, 5%, and 0.05%, respectively (Table 1).

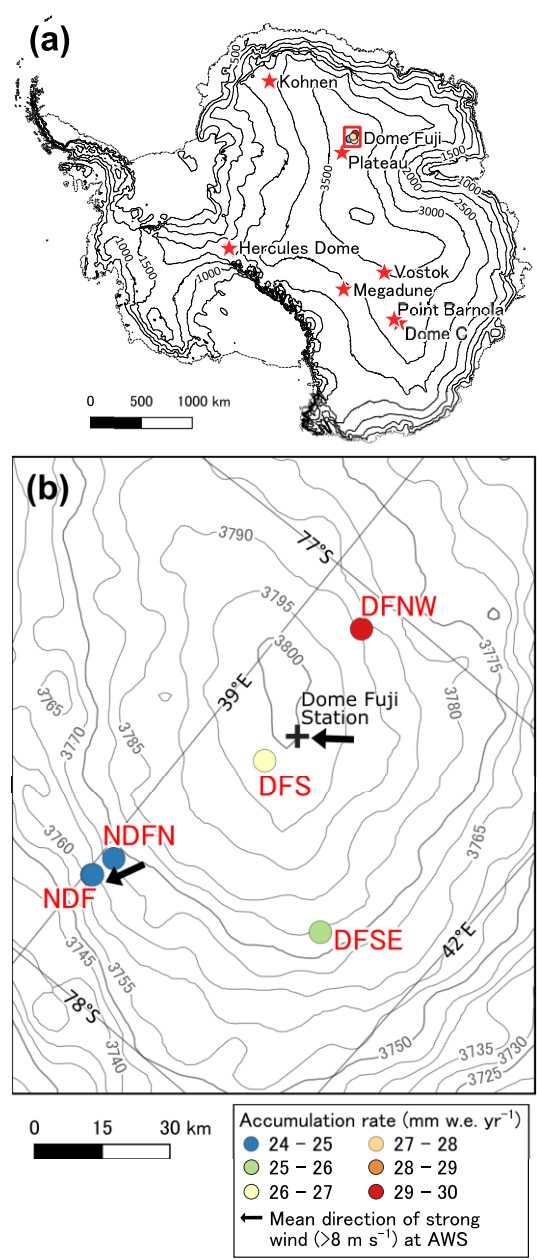


**Figure 1: Map of study sites. (a) Topographic map of Antarctica. The red rectangle indicates the Dome Fuji area. (b) Enlarged view of the Dome Fuji area. The solid circle marker indicates a site where one or two firn cores were collected for this study. The color of the marker represents the accumulation rate for 1980–2018 CE (Oyabu et al., 2023). Two black arrows show the mean direction of strong wind (>8 m s⁻¹) measured at the Automatic Weather Station (AWS) located in the Dome Fuji Station (53°) (Fujita et al., 2011)**

**and the NDF (26°) (https://ads.nipr.ac.jp/real-time-monitors/ndf/) (this study). Contours in (a) and (b) show elevation (m a.s.l.) based on the CryoSat-2-derived elevation model referenced to WGS84 (Helm et al., 2014).**



**Table 1: Information of the sampling sites and firn cores.**

| Site name | NDF18[a] | NDF13[a] | NDFN | DFSE | DFS10[a] | DFNW |
|---|---|---|---|---|---|---|
| Latitude (°) | −77.788 | −77.787 | −77.736 | −77.584 | −77.395 | −77.071 |
| Longitude (°) | 39.054 | 39.059 | 39.118 | 41.024 | 39.617 | 39.531 |
| Elevation[b] (m) | 3764 | 3764 | 3772 | 3780 | 3798 | 3789 |
| Slope[b] (‰) | 1.7 | 1.7 | 1.4 | 0.6 | 0.5 | 1.8 |
| Accumulation rate[c] (mm w.e. yr$^{-1}$) | 24.5 ± 1.4 | 23.7 ± 0.5 | 24.6 ± 0.1 | 25.1 ± 0.1 | 26.7 ± 0.7 | 29.4 ± 0.6 |
| 10 m snow temperature (°C) | −56.4 | (−56.4)[d] | (−56.4)[d] | −58.1 | (−57.3)[e] | −56.2 |
| 10 m wind speed[f] (m s$^{-1}$) | 5.0 | 5.0 | 5.0 | 4.8 | 5.0 | 5.1 |
| Insolation in December[g] (W m$^{-2}$) | 445.2 | 445.2 | 445.2 | 445.0 | 445.1 | 445.1 |
| Core length (m) | 152 | 31 | 142 | 41 | 122 | 43 |
| Sampling date | 19–27 Dec. 2017 | 24–25 Dec. 2012 | 14–29 Dec. 2018 | 31 Dec. 2017 − 2 Jan. 2018 | 15–20 Dec. 2010 | 5–7 Jan. 2018 |

[a]The number after the alphabet indicates the year of field campaign (Fujita et al., 2016; Oyabu et al., 2023).
[b]Based on the CryoSat-2-derived elevation model referenced to WGS84 (Helm et al., 2014).
[c]Average and standard deviation for 1885–1992 CE, derived from the depth-age relationships of firn cores (Oyabu et al., 2023).
[d]Data at the NDF18 site.
[e]Data at the Dome Fuji Station.
[f]Scalar average of hourly data for 1980–2018 from ERA5 reanalysis.
[g]Average for 1980–2018 from ERA5 reanalysis.


## 2.2 Measurements

The density, microstructural anisotropy, and SSA were measured, as described below (summarized in Table 2). All
measurements were performed at the National Institute of Polar Research (NIPR), Japan.





**Table 2: Characteristics of the measurements performed on the six firn cores.**

| ID | Method | Target properties | Spatial resolution (measurement intervals) | Shape and size of the sample[a] | Samples measured | Section described |
|---|---|---|---|---|---|---|
| 1 | Density measurement using the gamma-ray transmission method | Density ($\rho$) | 0.0033 m (0.003 m) | Slab $\Delta x = 0.5$ m, $\Delta y = 0.06$ m, $\Delta z = 0.038$ m[b] | NDF18, NDFN, DFSE, and DFNW | 2.2.1 |
| 2 | Dielectric tensor method (DTM) at 15–20 GHz at −30°C | Relative permittivity along the vertical ($\varepsilon_v$) and horizontal ($\varepsilon_h$), and dielectric anisotropy ($\Delta\varepsilon = \varepsilon_v - \varepsilon_h$) | ~0.038 m (0.02 m) | Same as ID1 | Same as ID1 | 2.2.2 |
| 3 | Dielectric tensor method (DTM) at 33–35 GHz at −16°C | Same as ID2 | ~0.022 m (0.005 m) | Slab $\Delta x = 0.5$ m, $\Delta y = 0.06$ m, $\Delta z = 0.005$ m | NDF13 and DFS10[c] | 2.2.2 |
| 4 | Near-infrared (NIR) reflectance measurement using an optical line scanning method | Specific surface area (SSA) | 0.0025 m | Same as ID1 | Same as ID1 | 2.2.3 |

[a]$\Delta x$, $\Delta y$, and $\Delta z$ are sample lengths, width, and thickness, respectively.
[b]$\Delta z$ of the NDFN core is 0.063 m for ID1 and 0.043 m for ID2 and ID4.
[c]Data from Fujita et al. (2016).


### 2.2.1 Density measurement using the gamma-ray transmission method

The high-resolution density ($\rho$) profiles were measured using a gamma-ray transmission densimeter (PH-1100N, Nanogray Inc., Japan) (Miyashita, 2008) on the NDF18, NDFN, DFSE, and DFNW cores (see Fig. 2a for the measurement setup). The densimeter measures the attenuation of a gamma-ray beam that passes through the firn samples and converts it into $\rho$ using

Beer's law. The firn cores were cut into a slab-shaped sample, typically 0.5 m long, 0.06 m wide, and 0.038 m thick. The cut surfaces of the samples were smoothed using a microtome for precise sample parallelism and thickness. We used a slit of lead with an aperture width of 0.0033 m to detect the gamma-ray transmission through samples, and we measured $\rho$ at 0.003 m depth increments. The error in $\rho$ is 1–2%, including the error of sample thickness and gamma-ray counting.






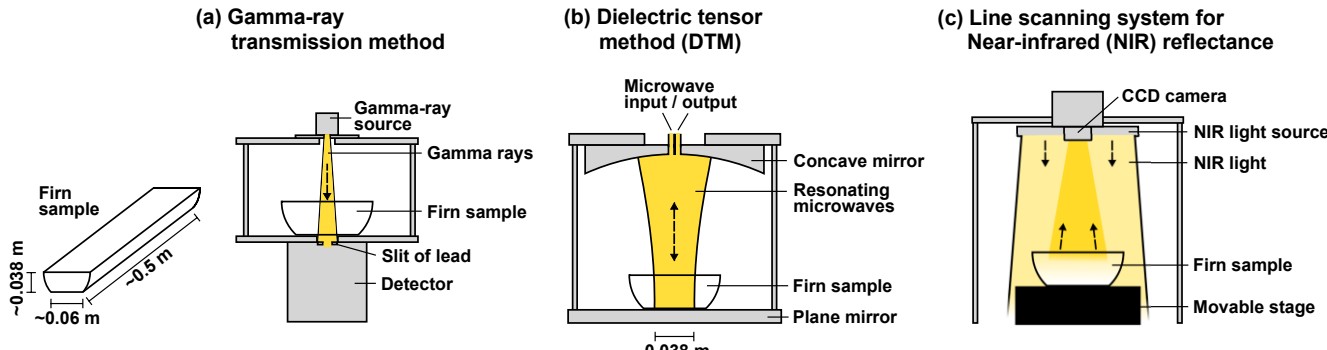

**Figure 2: Schematics of the physical measurement setups. (a) Gamma-ray transmission density measurement. (b) Dielectric tensor method (DTM) at 15–20 GHz (microwave) (in the case of the NDF18, NDFN, DFSE, and DFNW cores). (c) Line scanning system for NIR reflectance. The schematics illustrate the cross-section of the setups, along with the horizontal sections of firn cores. The firn**
**cores were moved perpendicular to the sheet during measurement.**

### 2.2.2 Dielectric tensor method

The relative permittivity of firn at high frequencies between short radio wave (MHz) and millimeter wave (GHz) frequencies
primarily depends on the firn density (e.g., Cumming, 1952; Kovacs et al., 1995; Fujita et al., 2014). The relative permittivity
(hereinafter, permittivity) is also influenced by the firn microstructure. If the firn has an anisotropic microstructure, that is,
comprising ice matrix and pore spaces with anisotropic shapes, the permittivity parallel to the anisotropic longer axis is larger
than that along the anisotropic shorter axis (e.g., Lytle and Jezek, 1994; Fujita et al., 2009, 2014, 2016; Leinss et al., 2016).

We detected firn density and microstructural anisotropy through the dielectric tensor method (DTM) (Fig. 2b) (Matsuoka et
al., 1998; Fujita et al., 2009, 2014, 2016; Saruya et al., 2022a, b). We used an open resonator to simultaneously measure the
permittivity parallel (vertical, $\varepsilon_v$) and perpendicular (horizontal, $\varepsilon_h$) to the core axis. Subsequently, we calculated their
difference $\Delta\varepsilon$ ($= \varepsilon_v - \varepsilon_h$), which we define as dielectric anisotropy, as a proxy for microstructural anisotropy. We performed
the DTM on the five firn cores: NDF13, NDF18, NDFN, DFSE, and DFNW. Additionally, we used the published data of the
DFS10 core (Fujita et al., 2016). The NDF18, NDFN, DFSE, and DFNW cores were measured at 15–20 GHz frequencies at
−30°C in 2019–2021. The Gaussian beam through a sample had a diameter of ~0.038 m at the $1/e^2$ intensity level of peak
intensity, and we measured $\varepsilon$ in 0.02 m depth increments. The samples measured were the same as those for the gamma-ray
transmission density measurement. The NDF13 and DFS10 cores were measured at 33–35 GHz at −16°C in 2012–2015. The
beam diameter and depth increment were ~0.022 and 0.005 m, respectively. The slab-shaped samples were ~0.005 m thick
and 0.06 m wide.



The permittivity ($\varepsilon_\mathrm{h}$) was converted to density ($\rho_\varepsilon$) using empirical relationships between $\varepsilon_\mathrm{h}$ and $\rho_\varepsilon$ at the measurement temperature of −16°C or −30°C. For measurements at −30°C (NDF18, NDFN, DFSE, and DFNW), we used the following empirical equation derived by Oyabu et al. (2023):

$$\rho_\varepsilon = -20.15\varepsilon_\mathrm{h}^3 + 99.801\varepsilon_\mathrm{h}^2 + 243.02\varepsilon_\mathrm{h} - 220.57 \tag{1}$$

For measurements at −16°C (NDF13 and DFS10), Oyabu et al. (2023) derived the following equation from a relationship between $\varepsilon_\mathrm{h}$ and $\rho_\varepsilon$ given by Fujita et al. (2014):

$$\rho_\varepsilon = -5.556\varepsilon_\mathrm{h}^3 - 4.0922\varepsilon_\mathrm{h}^2 + 494.14\varepsilon_\mathrm{h} - 436.121 \tag{2}$$

The analytical uncertainty of $\varepsilon_\mathrm{h}$ is ± 0.005, and the total uncertainty in the converted density is ~6–14 kg m$^{-3}$, including errors in $\varepsilon$ and density for calibration (Oyabu et al., 2023).

Following Fujita et al. (2009), $\Delta\varepsilon$ is related to the axial ratio $R$, the ratio between the correlation length of pore space in the vertical direction and that in the horizontal direction. It is expressed using the following equation:

$$R = 3.761\Delta\varepsilon + 0.954. \tag{3}$$

The analytical uncertainty of $\Delta\varepsilon$ is <0.001 (Fujita et al., 2016; Saruya et al., 2022b), and the total uncertainty in $R$ is 0.08, including the errors in $\Delta\varepsilon$ and $R$ for calibration.

### 2.2.3 SSA

Snow SSA is the area of ice-to-pore interface per unit mass of firn (e.g., Legagneux et al., 2002). Assuming that snow grains are spherical, the relationship between SSA and sphere radius ($r$) is expressed as follows (e.g., Gallet et al., 2009):

$$\mathrm{SSA} = \frac{3}{r\rho_\mathrm{ice}} \tag{4}$$

where $\rho_\mathrm{ice}$ is the density of pure ice (917 kg m$^{-3}$ at 0°C). The SSA can be estimated by measuring the reflectance of NIR light with a wavelength of 750–1400 nm (e.g., Matzl and Schneebeli, 2006; Gallet et al., 2009, 2011; Libois et al., 2015), based on the Mie theory (e.g., Wiscombe and Warren, 1980).

We used an optical line scanning system (e.g., Takata et al., 2004) (Fig. 2c) to measure the NIR reflectance at high resolution (0.0025 m) on the NDF18, NDFN, DFSE, and DFNW cores. The system comprises LED light sources with a center wavelength of 930 nm and a half power width of ~50 nm, along with a charge-coupled device (CCD) camera (C3077-79, HAMAMATSU, Japan) with high sensitivity in 800–1100 nm, mounted 0.4 m above the measured surface of a firn sample. A slab-shaped sample, whose surface was smoothed using a microtome, was placed on a movable stage. During the measurement, the sample was irradiated with NIR light while moving the stage at a constant speed, and the CCD camera continuously recorded the brightness of pixels in the direction perpendicular to the moving direction of the sample stage. The individual pixel data were





compiled into an image of the entire firn sample. The brightness was calibrated into reflectance using four reflectance standards: 98.8%, 81.9%, 67.7%, and 23.5%. Subsequently, we converted the reflectance into SSA using an empirical

relationship between the reflectance measured by our system and the SSA obtained from hemispherical NIR reflectance (see Appendix A for details). Our system has a measurement resolution of 0.00025 m, and we obtained the data at 0.0025 m depth increments. The error in SSA is ~15%, including the error of SSA obtained from hemispherical NIR reflectance and the regression curve for calibration (Appendix A).

**2.3 Data processing and analyses**

We standardized the depth resolutions of the measured properties ($\rho$, $\varepsilon_{v}$, $\varepsilon_{h}$, $\Delta\varepsilon$, and SSA) and unified the depth intervals using the following procedures. First, we manually removed outliers at the core breaks from the raw data. Then, we smoothed the $\rho$ and SSA data by 7- and 8-point moving averages, respectively, both corresponding to a resolution of ~0.02 m. For the $\varepsilon$ and $\Delta\varepsilon$ profiles, the NDF13 and DFS10 data (measured at 33–35 GHz) were smoothed by a 4-point moving average (resolution of

~0.035 m). No smoothing was applied to the NDF18, NDFN, DFSE, and DFNW core data (measured at 15–20 GHz) with a measurement resolution of ~0.038 m. Finally, we linearly interpolated all data at the 0.02 m depth intervals except for the core breaks. The intervals used for interpolation did not affect our result, as long as they were smaller than ~0.05 m to resolve firn layers in our cores.

To identify the general trends in $\rho$, $\varepsilon_{h}$, $\Delta\varepsilon$, and SSA, we calculated 0.5 m moving averages of their interpolated data at 0.02 m intervals. The average was not calculated for the 0.5 m interval containing nine data points or less (<0.1 m) because the data often represent only one layer. To analyze the firn layering, we detrended the $\rho$, $\Delta\varepsilon$, and SSA data by subtracting their 0.5 m moving averages from the 0.02 m interpolated data. We calculated 1 m moving standard deviations (S.D.) of the detrended data as a measure of variability in $\rho$, $\Delta\varepsilon$, and SSA, and linear correlation coefficients between the pairs of properties within 1

m moving intervals to identify the depth ranges with their strong and weak covariations. The moving S.D. and correlation coefficients were only calculated for the NDF18, NDFN, DFSE, and DFNW cores with the $\rho$ and SSA data. The 1 m intervals containing a data coverage of <0.5 m were excluded from the analyses.

**3 Results**

We describe the general trend of density ($\rho$ or $\rho_{\varepsilon}$), $\Delta\varepsilon$, and SSA and their variabilities in the top 10 m of firn at each site. We focus on the top few meters, where initial metamorphism is expected to be large due to large vertical TGs (e.g., Azuma and others, 1997). Figure 3a–d show $\rho$, $\varepsilon_{v}$, $\varepsilon_{h}$, $\Delta\varepsilon$, and SSA of the NDF18 core (see Fig. 3e–t for the data of other cores). The 0.5



m moving averages of $\rho$, $\varepsilon_\mathrm{h}$, $\Delta\varepsilon$, and SSA are shown (black dashed lines in Fig. 3) to identify their general trends and variabilities around the trends.


## 3.1 Density

The density profiles of the six cores around Dome Fuji are characterized by relatively large variability without significant trends in the top 4 m (Fig. 3), contrary to the expectation from typical firn densification processes such as grain packing and sintering. For the NDF18 core, $\rho$ (and $\rho_\varepsilon$) ranges between 255 and 460 kg m$^{-3}$ in 0–4 m, and the slopes of linear regression for

the data are $-2$ and $-4$ kg m$^{-4}$ (i.e., kg m$^{-3}$ per meter depth) for 0–2 and 2–4 m, respectively ($p > 0.05$). Below 4 m depth, significant linear trends of 12, 13, and 15 kg m$^{-4}$ are observed for 4–6, 6–8, and 8–10 m, respectively ($p < 0.05$). The 0.5 m moving averages of $\rho_\varepsilon$ measured in all six cores agree with each other within the measurement error of ~6–14 kg m$^{-3}$ (Fig. 4a). Similar to the NDF18 core, the NDFN, DFSE, and DFNW cores do not show significant increasing trends of $\rho_\varepsilon$ in the top 2 m (the NDF13 and DFS10 cores were excluded from the analysis due to sparse data). Below ~2 m, the moving averages of $\rho_\varepsilon$

increase to ~365 kg m$^{-3}$ at ~4 m and exceed the $\rho_\varepsilon$ range in the top ~2 m.

The density in each core fluctuates significantly around the moving average on a scale of ~0.1 m, reflecting the density layering of firn (Fig. 3), whose amplitudes appear to decrease with depth. To investigate the evolution of the density variability, we calculated the 1 m moving S.D. of the deviations of the 0.02 m resolution data from the 0.5 m moving average for the NDF18,

NDFN, DFSE, and DFNW cores (Fig. 6a, see Fig. 5 for the deviations $\Delta\rho$). The moving S.D. of $\Delta\rho$ decrease from ~40 kg m$^{-3}$ at 0.75 m to ~20 kg m$^{-3}$ at 10 m in the four cores; a decreasing trend is typically observed in polar firn (e.g., Fujita et al., 2009, 2016; Hörhold et al., 2011). In addition, the moving S.D. fluctuates by ~20 kg m$^{-3}$, with large values at the depths with high-density, e.g., at around 5.4 m in the NDF18 core (Fig. 5a) and 5.7 m in the DFSE core (Fig. 5c). The local maxima and minima of the moving S.D. for the four cores tend to appear at similar depths (maxima around 2.5, 5.5, and 8.5 m depths and minima

around 4.0 and 7.0 m depths).









**Figure 3: Measured physical properties for 0–10 m of the six firn cores within 60 km around Dome Fuji. (a, e, i, m) Density $\rho$, (b, f, j, n, q, s) relative permittivity parallel to the core axis $\varepsilon_v$ and perpendicular to the core axis $\varepsilon_h$, (c, g, k, o, r, t) dielectric anisotropy $\Delta\varepsilon$ (= $\varepsilon_v - \varepsilon_h$), and (d, h, l, p) SSA. Gray, dark solid, and dashed lines indicate raw data for $\rho$ and SSA, data at 0.02 m intervals, and 0.5 m moving average, respectively. The axis of $\rho_\varepsilon$ next to the $\varepsilon$ axis is obtained from the relationship between $\varepsilon_h$ and $\rho_\varepsilon$ (Eq. 1 and 2) (Oyabu et al., 2023). The axis of the axial ratio next to the $\Delta\varepsilon$ axis is obtained from the empirical relationship between the axial ratio and $\Delta\varepsilon$ (Eq. 3) (Fujita et al., 2009). The axis of grain radius $r$ next to the SSA axis is obtained from SSA (Eq. 4).**



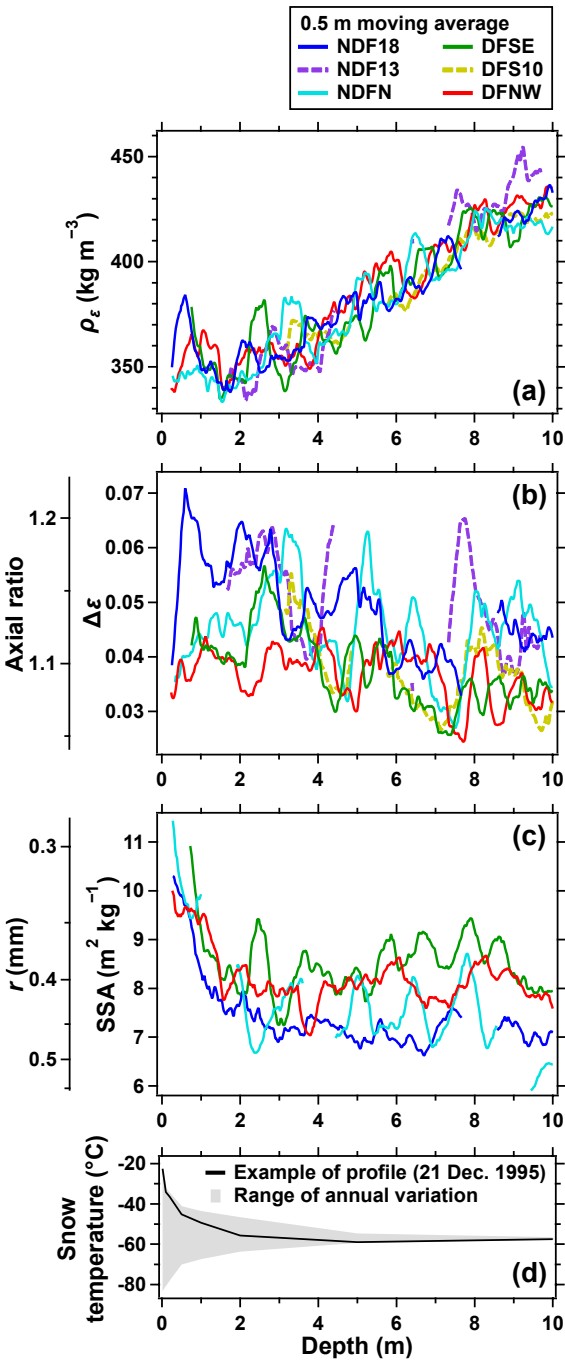

**Figure 4: General trends in $\rho_\varepsilon$, $\Delta\varepsilon$, and SSA of the NDF18, NDF54, NDFN, DFSE, DFS10, and DFNW cores. (a) Moving average of $\rho_\varepsilon$ using a 0.5 m window. (b, c) Same as (a) but for $\Delta\varepsilon$ and SSA, respectively. (d) Snow temperature profile measured in a borehole at the Dome Fuji Station (Azuma and others, 1997). The black line is an example profile when TG is high near the surface (12:00 on 21 December 1995). The shading indicates the range of the snow temperature during 1995.**





**Figure 5: Anomalies (referred to as Δ) of ρ, Δε, and SSA of the (a) NDF18, (b) NDFN, (c) DFSE, and (d) DFNW cores. The Δρ, Δ(Δε),**
**and ΔSSA are obtained by subtracting the 0.5 m moving average from their data at 0.02 m intervals. Amplitudes of Δρ and ΔSSA**
**decrease with depth, while the amplitude of Δ(Δε) is low at the shallowest depths and increases toward approximately 5 m. Prominent**
**maxima (or minima) of Δρ, Δ(Δε), and ΔSSA appear at similar depths.**



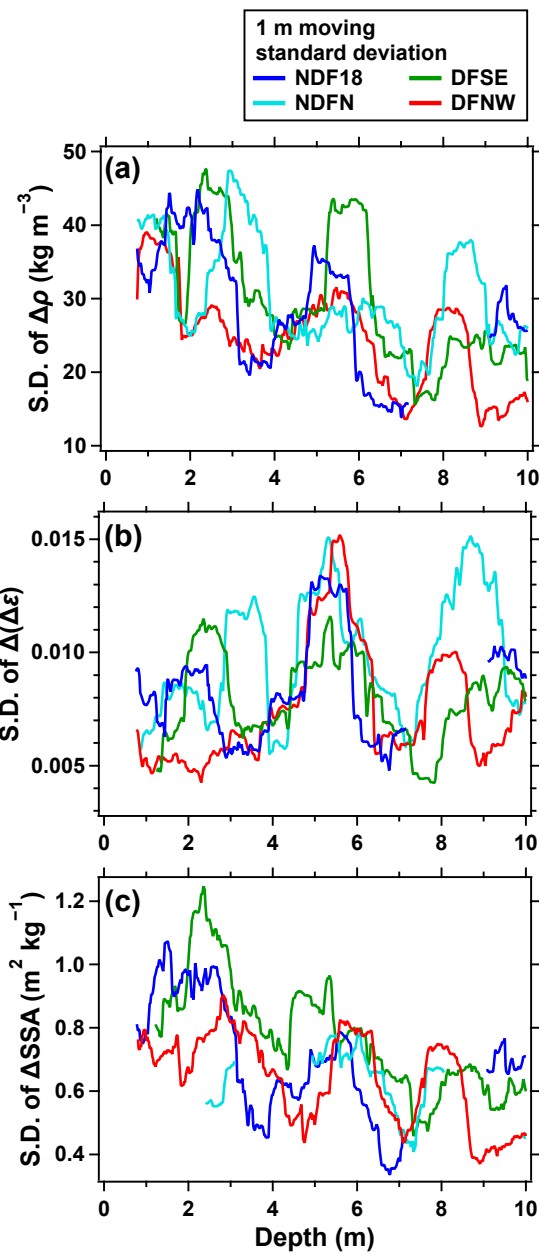

**Figure 6: Standard deviations (S.D.) of $\Delta\rho$, $\Delta(\Delta\varepsilon)$, and $\Delta$SSA of the NDF18, NDFN, DFSE, and DFNW cores. (a) Moving S.D. of $\Delta\rho$ using a 1 m window. Windows with a data coverage of <0.5 m were excluded from the calculation. (b, c) Same as (a) but for $\Delta(\Delta\varepsilon)$ and $\Delta$SSA, respectively.**






## 3.2 Dielectric anisotropy

The $\Delta\varepsilon$ of the NDF18 core increases rapidly with depth from ~0.035 near the surface to ~0.07 at 0.5 m, corresponding to the increasing axial ratio from ~1.1 to ~1.2, indicating the development of the vertically elongated structure of ice matrix and pores (Fig. 3c). Below 0.5 m depth, the moving average of $\Delta\varepsilon$ generally decreases with depth with fluctuations on the scales from

several tens of centimeters to meters (e.g., maxima around 2.8 m depth and minima around 3.2 m depth). The initial increase and subsequent general decrease of the moving average of $\Delta\varepsilon$ are also observed at other sites around Dome Fuji (Fig. 3g, k, o, r, t and Fig. 4b). Similar development in microstructural anisotropy near the surface has been observed at high-accumulation sites: Hercules Dome in West Antarctica (160–200 mm w.e. yr$^{-1}$) (Hörhold et al., 2009) and Summit in Greenland (220 mm w.e. yr$^{-1}$) (Lomonaco et al., 2011), but not at low-accumulation sites in East Antarctica (<30 mm w.e. yr$^{-1}$) probably due to

poor core quality or recovery rate in the top several meters in the previous studies (Fujita et al., 2009, 2016). The average $\Delta\varepsilon$ for 0.0–0.5 m depth ranges from 0.03 to 0.04 for most of our studied sites. Below 0.5 m depth, $\Delta\varepsilon$ values are generally higher at the southern sites (NDF and NDFN) than at the northern sites (DFSE, DFS, and DFNW).

The deviations of $\Delta\varepsilon$ from the 0.5 m moving averages, $\Delta(\Delta\varepsilon)$, show large variations on a scale of ~0.1 m, indicating the

microstructural layering of the firn (Fig. 5). The evolution of $\Delta\varepsilon$ variability is represented by the 1 m moving S.D. of $\Delta(\Delta\varepsilon)$ for the NDF18, NDFN, DFSE, and DFNW cores (Fig. 6b), which are relatively low at the shallowest depths and increase with depth toward the first local maxima between 2 and 4 m. Below 4 m, the moving S.D. for each core fluctuates with amplitudes of >0.005, and their maxima occur at similar depths (around 5.5 and 8.5 m). The maxima and minima in the moving S.D. of $\Delta(\Delta\varepsilon)$ appear at similar depths as those in the moving S.D. of $\Delta\rho$ (Fig. 6).


Close inspection of the $\Delta(\Delta\varepsilon)$ from the four cores reveals that their prominent maxima (minima) coincide with those of $\Delta\rho$ (Fig. 5), e.g., at 4.4 and 5.4 (2.0 and 6.0) m in the NDF18 core (Fig. 5a). To investigate the relationship between $\Delta(\Delta\varepsilon)$ and $\Delta\rho$, we calculated their correlation coefficients with 1 m moving window (Fig. 7a). $\Delta(\Delta\varepsilon)$ and $\Delta\rho$ positively correlated for all cores throughout the 0–10 m depth range. This result is consistent with the previous finding that high-density layers show high

$\Delta\varepsilon$ (Fujita et al., 2009, 2016). The correlation coefficient is the smallest near the surface and increases until ~3 m, probably because the variability of $\Delta(\Delta\varepsilon)$ is small near the surface (for example, large $\Delta\rho$ peaks at ~1.4 m of the NDF18 core and ~1.2 m of the NDFN, DFSE, and DFNW cores are not accompanied by marked peaks in $\Delta(\Delta\varepsilon)$).

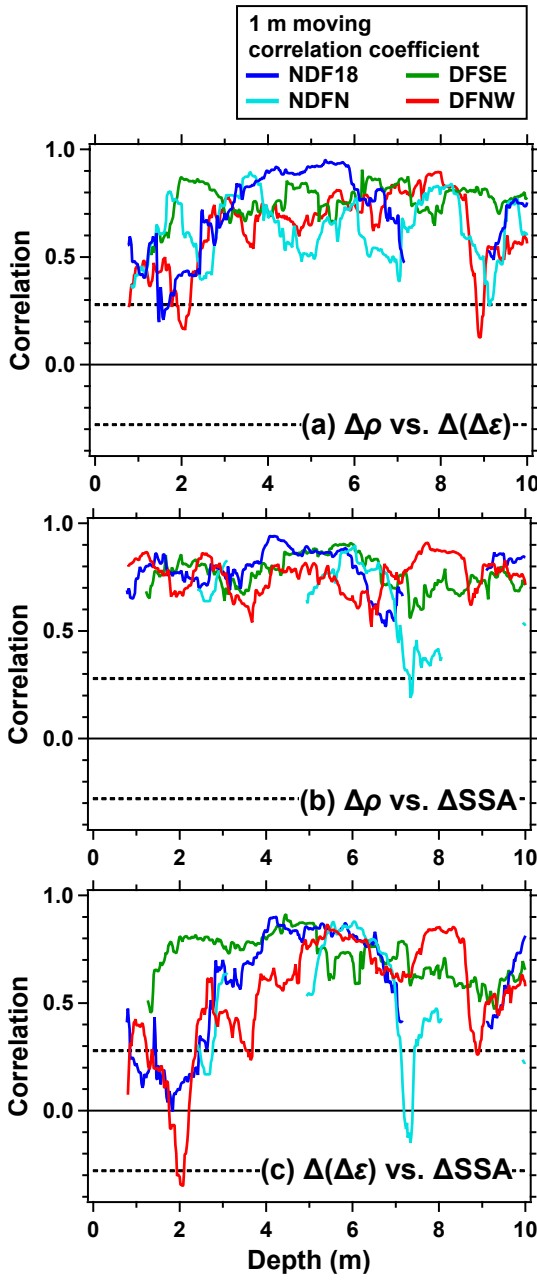

**Figure 7: Correlation coefficients between the pairs of Δρ, Δ(Δε), and ΔSSA of the NDF18, NDFN, DFSE, and DFNW cores. (a) The moving correlation coefficient between Δρ and Δ(Δε) using a 1 m window. Windows with a data coverage of <0.5 m were excluded from the calculation. The dotted horizontal lines indicate the 95% confidence level of the correlation coefficient (data between the two lines are not significant, p > 0.05). (b, c) Same as (a) but for the correlation coefficient between Δρ and ΔSSA and between Δ(Δε) and ΔSSA, respectively.**





### 3.3 SSA

The SSA of the NDF18 core decreases from ~11 m² kg⁻¹ at the surface to ~7 m² kg⁻¹ at ~3 m, corresponding to an increase in
$r$ from ~0.3 mm to ~0.5 mm (Fig. 3d). The rate of decrease becomes smaller with depth, and the 0.5 m moving average of SSA
is stable below ~3 m. The initial decrease and relatively stable values of SSA below ~3 m are also observed at NDFN, DFSE,
and DFNW (Fig. 3h, l, p and Fig. 4c). The SSA decrease in the top few meters is consistent with the observations at Point
Barnola (Calonne et al., 2017), Dome C (Gallet et al., 2011), Kohnen Station, and Hercules Dome (Linow et al., 2012).
Furthermore, the southern sites (NDF and NDFN) show smaller SSA than the other sites (DFSE and DFNW). Around the
moving averages, the SSA fluctuates on a scale of ~0.1 m (Fig. 3d, h, l, p). The 1 m moving S.D. of ΔSSA (deviation of SSA
from its 0.5 m moving average, Fig. 5) decrease with depth in the NDF18, NDFN, DFSE, and DFNW cores (Fig. 6c). The
maxima and minima of the moving S.D. occur at similar depths in the four cores, coinciding with those of the moving S.D. of
$\Delta\rho$ and $\Delta(\Delta\varepsilon)$ (maxima around 2.5, 5.5, and 8.5 m and minima around 4.0 and 7.0 m) (Fig. 6).

The evolutions of the relationships of SSA with $\rho$ and $\Delta\varepsilon$ are investigated by scatter plots of SSA against $\rho$ for the four cores,
in which colors of the markers represent the depths (Fig. 8a, c, e, g) or $\Delta\varepsilon$ (Fig. 8b, d, f, h), as well as the correlation coefficients
of ΔSSA to $\Delta(\Delta\varepsilon)$ or $\Delta\rho$ with 1 m moving window (Fig. 7b and 7c). The prominent maxima and minima of ΔSSA tend to
coincide with those of $\Delta\rho$ and $\Delta(\Delta\varepsilon)$; for example, the high (low) SSA layer at 5.4 (6.0) m shows high (low) $\Delta\rho$ and $\Delta(\Delta\varepsilon)$ in
the NDF18 core (Fig. 5a). The 1 m moving correlation coefficients between the $\Delta\rho$ and ΔSSA (Fig. 7b) and between $\Delta(\Delta\varepsilon)$
and ΔSSA (Fig. 7c) are predominantly positive for the four cores (see also Fig. 8). In addition, while the correlation coefficient
between $\Delta\rho$ and ΔSSA is consistently high, the correlation coefficients between $\Delta(\Delta\varepsilon)$ and ΔSSA (or $\Delta\rho$) is small or
insignificant near the surface and increases until ~3 m. This is partly due to the absence of prominent $\Delta(\Delta\varepsilon)$ maxima at the
depths of ΔSSA and $\Delta\rho$ maxima near the surface.






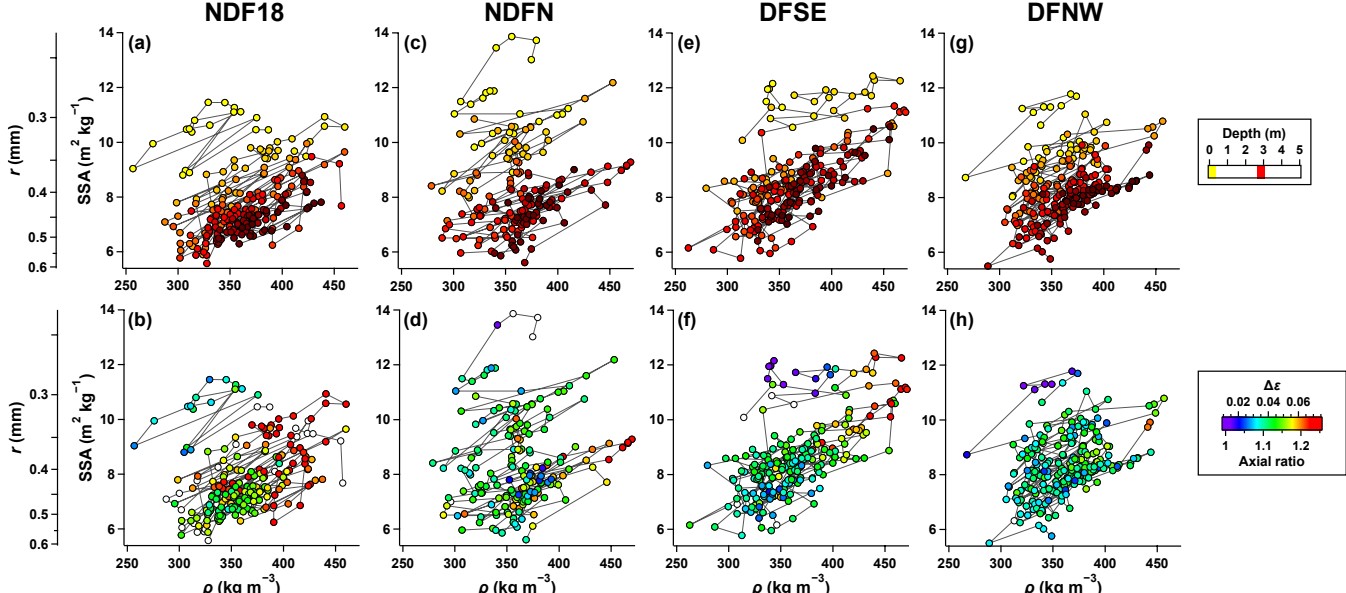

**Figure 8: Relationship between $\rho$, $\Delta\varepsilon$, and SSA of the NDF18, NDFN, DFSE, and DFNW cores. (a, c, e, g) Scatter plot of SSA against $\rho$ for 0–5 m with colors of the markers representing the depths. Lines connect the symbols in the depth order. (b, d, f, h) Same as (a, c, e, g), but colors represent $\Delta\varepsilon$. The white color indicates no data of $\Delta\varepsilon$. Variations of $\rho$ and SSA on the scale of ~0.1 m are positively**

**correlated. $\Delta\varepsilon$ increases within the top ~3 m, with more significant increases at depths with higher $\rho$ and SSA. $\Delta\varepsilon$ is highest at NDF18 and lowest at DFNW.**

## 4 Discussion

Our continuous and high-resolution data of $\rho$, $\varepsilon$, $\Delta\varepsilon$, and SSA from six firn cores around Dome Fuji provide detailed views of

the evolution of these properties at low-accumulation sites (<30 mm w.e. yr$^{-1}$). In the following, we discuss the key processes for the evolution of density (Sect. 4.1), $\Delta\varepsilon$ (Sect. 4.2), and SSA (Sect. 4.3) in the top few meters that are common to the six cores. Further, we discuss the causes of spatial and temporal variabilities in these properties around Dome Fuji.

### 4.1 Density evolution

**4.1.1 Processes for the density evolution**

Generally, the density of near-surface firn is expected to increase with depth due to settling of snow grains under overburden pressure. However, our $\rho$ (or $\rho_\varepsilon$) data in the NDF18, NDFN, DFSE, and DFNW cores do not show significant increase in the ~0–4 m range (Fig. 4a). The lack of density increase in the four cores is consistent with previously published data from pit studies near the respective drilling sites (Fig. 9 and Table 3). Previous studies at other low-accumulation sites at the altitudes



above 3000 m a.s.l. have also shown the lack of significant density increase in the top 2 or 3 m, e.g., at Point Barnola (Calonne et al., 2017) and Dronning Maud Land plateau (Endo and Fujiwara, 1973; Weinhart et al., 2020). These observations suggest that the lack of density increase in the top few meters is a common feature in the Antarctic inland plateau.

Endo and Fujiwara (1973) and Calonne et al. (2017) speculated that the lack of mean density increase in near-surface firn is
due to insufficient overburden pressure to facilitate the settling of firn with high viscosity and resistance to deformation due to low temperature (e.g., Kojima, 1971; Vionnet et al., 2012). However, slow densification due to the slight overburden pressure cannot explain the observed density decrease for the top ~2 m (Fig. 3a and Table 3). A potential process that may contribute to the density decrease is mass loss of the near-surface firn through TGM; because the near-surface firn is exposed to high TG, which facilitates vapor transports, part of upward vapor flux may be released to the atmosphere without condensing in the
shallower firn, potentially leading to a net mass loss. The minimum density observed at ~1.5 m (Fig. 3a) may suggest that the mass loss is particularly effective around this depth, where the vapor transport should be facilitated by the combination of strong TG (Fig. 3d) and high air permeability along the vertical direction with more vertically-elongated and coarse microstructure than the shallower depths (Fig. 3b and 3c) (e.g., Hörhold et al., 2009). Another possibility is that the density fluctuations in the top few meters reflect the temporal variations in surface density over the past few decades. To quantitatively
assess the effects of slight overburden pressure, mass loss processes, and past surface density changes on the density evolution, the observations of strain rate of the near-surface firn, mass balance by sublimation and condensation in the top few meters, and long-term trend of surface density would be necessary.

The density variability in the NDF18, NDFN, DFSE, and DFNW cores decreases with depth over the top 10 m (see moving
S.D. in Fig. 6a), consistent with previous observations of polar firn (e.g., Fujita et al., 2009, 2016; Hörhold et al., 2011). The earlier studies have suggested that the decrease in density variability is due to density dependence of densification rate; the densification rate of a high-density layer may be lower than that of a low-density layer because the vertical bonds between finer grains in high-density layers provide resistance to deformation (Alley et al., 1982; Freitag et al., 2004; Fujita et al., 2009, 2016). In contrast, a low-density layer, typically composed of coarser grains, may settle rapidly because its larger pores are
less likely to be filled with ice bonds during TGM. Our data are consistent with the previous studies, showing that the high-density layers (positive $\Delta\rho$) have more vertically elongated structure (positive $\Delta(\Delta\varepsilon)$) and finer grains (positive SSA) than the low-density layers (Fig. 5, 7, and 8) (Fujita et al., 2009). According to Fujita et al. (2009), the two types are referred to as initially high-density firn (IHDF) and initially low-density firn (ILDF), respectively.






**Figure 9:** Density profiles from the sites around Dome Fuji listed in Table 3. The red dashed line is the reference line of 350 kg m⁻³. The densities in the top 1 m in the NDF, DFNW, and DFNW regions and at DFSE are higher than those in the Dome region and at DK018. In the former regions, density do not significantly increase in 0–4 m, while densities in the latter regions do.





**Table 3: Near-surface density around Dome Fuji.**

| Site name | Latitude (°) | Longitude (°) | Sampling date | Pit/ Core | Depth (m) | Average (kg m⁻³)ᵃ | | | | Slope (kg m⁻⁴)ᵃ | | Subregion | Reference |
|---|---|---|---|---|---|---|---|---|---|---|---|---|---|
| | | | | | | 0–1 m | 1–2 m | 2–3 m | 3–4 m | 0–2 m | 2–4 m | | |
| NDF13 | −77.787 | 39.059 | 22 Dec. 2012 | Pit | 2.18 | 344 | 348 | | | −2 | | NDF | Oyabu et al. (2023) |
| NDF13 | −77.787 | 39.059 | 24–25 Dec. 2012 | Core, $\rho_\varepsilon$ | 31 | 344 | 338 | 350 | 353 | | 6 | NDF | This study |
| NDF18 | −77.788 | 39.054 | 26–28 Dec. 2017 | Pit | 4.02 | 378 | 350 | 363 | 366 | −17 | 4 | NDF | Oyabu et al. (2023) |
| NDF18 | −77.788 | 39.054 | 19–27 Dec. 2017 | Core, $\rho_\varepsilon$ | 152 | 359 | 345 | 358 | 362 | −10 | 5 | NDF | This study |
| NDF18 | −77.788 | 39.054 | 19–27 Dec. 2017 | Core, $\rho$ | 152 | 359 | 346 | 368 | 361 | −2 | −4 | NDF | This study |
| NDFN | −77.736 | 39.118 | 14–29 Dec. 2018 | Core, $\rho_\varepsilon$ | 142 | 346 | 342 | 354 | 371 | −4 | 18 | NDF | This study |
| NDFN | −77.736 | 39.118 | 14–29 Dec. 2018 | Core, $\rho$ | 142 | 348 | 340 | 347 | 374 | −6 | 31 | NDF | This study |
| DFSE | −77.584 | 41.024 | 31 Dec. 2017 –2 Jan. 2018 | Core, $\rho_\varepsilon$ | 41 | 371 | 347 | 361 | 357 | −13 | 7 | DFSE | This study |
| DFSE | −77.584 | 41.024 | 31 Dec. 2017 –2 Jan. 2018 | Core, $\rho$ | 41 | 372 | 350 | 357 | 366 | −14 | 19 | DFSE | This study |
| DFS10 | −77.395 | 39.617 | 22 Jan. 2010 | Pit | 2.3 | 313 | 326 | | | 12 | | Dome | Oyabu et al. (2023) |
| DFS10 | −77.395 | 39.617 | 15–20 Dec. 2010 | Core, $\rho_\varepsilon$ | 122 | | | | 364 | | 6 | Dome | Fujita et al. (2016) |
| DF1997a | −77.373 | 39.614 | 18 Jan. 1997 | Pit | 3.68 | 311 | 333 | 364 | 383 | 13 | 21 | Dome | Oyabu et al. (2023) |
| DF1997a | −77.373 | 39.614 | 22 Feb. 1997 | Pit | 1.2 | 323 | | | | | | Dome | Oyabu et al. (2023) |
| DF1997a | −77.373 | 39.614 | 5 Mar. 1997 | Pit | 1.36 | 317 | | | | | | Dome | Oyabu et al. (2023) |
| DF1997a | −77.373 | 39.614 | 4 Apr. 1997 | Pit | 2.16 | 321 | 353 | | | 33 | | Dome | Oyabu et al. (2023) |
| DF1997a | −77.373 | 39.614 | 5 May. 1997 | Pit | 1.27 | 321 | | | | | | Dome | Oyabu et al. (2023) |
| DF1997a | −77.373 | 39.614 | 5 Aug. 1997 | Pit | 1.12 | 321 | | | | | | Dome | Oyabu et al. (2023) |
| DF1997a | −77.373 | 39.614 | 10 Sep. 1997 | Pit | 1.13 | 344 | | | | | | Dome | Oyabu et al. (2023) |
| DF1997a | −77.373 | 39.614 | 4 Oct. 1997 | Pit | 1.34 | 352 | | | | | | Dome | Oyabu et al. (2023) |
| DF1997a | −77.373 | 39.614 | 18 Nov. 1997 | Pit | 1.1 | 337 | | | | | | Dome | Oyabu et al. (2023) |
| DF1997a | −77.373 | 39.614 | 26 Dec. 1997 | Pit | 2.3 | 347 | 371 | | | 30 | | Dome | Oyabu et al. (2023) |
| DF1997a | −77.373 | 39.614 | 4 Feb. 2003 | Pit | 3.8 | 378 | 379 | 439 | 463 | 18 | 22 | Dome | Oyabu et al. (2023) |
| DF Station | −77.318 | 39.704 | 26–28 Nov. 2003 | Pit | 3 | 323 | 356 | 354 | | 33 | | Dome | K. Fujita, unpublished data |
| MD732 | −77.298 | 39.786 | 10–11 Dec. 2007 | Pit | 4.02 | 331 | 354 | 376 | 394 | 28 | 10 | Dome | Hoshina et al. (2014) |
| DF1997b | −77.000 | 39.583 | 25 Nov. 1997 | Pit | 1.85 | 338 | 360 | | | 19 | | DFNW | Oyabu et al. (2023) |
| DFNW | −77.071 | 39.531 | 5–7 Jan. 2018 | Core, $\rho_\varepsilon$ | 43 | 351 | 351 | 359 | 352 | 2 | −4 | DFNW | This study |
| DFNW | −77.071 | 39.531 | 5–7 Jan. 2018 | Core, $\rho$ | 43 | 352 | 347 | 362 | 352 | 2 | −4 | DFNW | This study |
| MD708 | −77.096 | 39.911 | 8 Dec. 2007 | Pit | 1.02 | 377 | | | | | | DFNW | Sugiyama et al. (2012) |
| DK018 | −77.171 | 38.569 | 16 Dec. 2007 | Pit | 1.02 | 332 | | | | | | DK018 | Sugiyama et al. (2012) |

ᵃDepth intervals with a data coverage of <0.5 m were excluded from the calculation.

Furthermore, the S.D. of $\Delta\rho$ in all four cores show three maxima at similar depths (Fig. 6a). Because the density variability may be predominantly determined by the occurrence and intensity of IHDF (see Fig. 5), the S.D. fluctuations may reflect the

past environmental conditions that form IHDFs. The intense IHDFs are formed by wind-packing (Koerner, 1971; Fujita et al., 2009), and their densities depend on wind speed (e.g., Sugiyama et al., 2012). Thus, wind is probably a key environmental factor controlling density variability (Fig. 6a). Accumulation rate and temperature may also determine density variability. Previous observations at multiple polar sites have shown a negative correlation between density variability in the top several meters and accumulation rate (temperature) for 25–180 mm w.e. yr⁻¹ (−53–−18 °C) (Hörhold et al., 2011), although the cause

for the correlation is not well understood. In our data, a spatial relationship between the density variability and accumulation rate is not observed (Fig. 6a and Table 1), presumably because of the narrow ranges of mean accumulation rate (23–30 mm w.e. yr⁻¹) and temperature (−56.2–−58.1 °C). Thus, we suggest that accumulation rate and temperature do not explain the density variability around Dome Fuji.






### 4.1.2 Variability of density evolution around Dome Fuji

The densities of the NDF, NDFN, DFSE, and DFNW cores agree with the published data from pit studies near the respective drilling sites, and the values in the top 1 m (~355 kg m$^{-3}$) are higher than those near the flat dome summit (central part of the dome summit and along the ridge: dome region and DK018 in Table 3) (~330 kg m$^{-3}$). Previous studies at the Dronning Maud
Land plateau above 3000 m a.s.l. (excluding the dome summit) have observed similar mean density in the top 1 m (~350–355 kg m$^{-3}$) (Endo and Fujiwara, 1973; Shiraiwa et al., 1996; Sugiyama et al., 2012; Weinhart et al., 2020). Shiraiwa et al. (1996) noted that the average density decreases toward the dome summit. Published pit studies at Dome C also reported low density at the surface (~280 kg m$^{-3}$), increasing in the top several tens of centimeters (Gallet et al., 2011), similar to the density profiles near the Dome Fuji summit (Fig. 9). Our results and previous observations suggest that relatively low density near the surface
is a local characteristic at the dome summit over the vast inland plateau.

To further investigate the density variability around Dome Fuji, we categorized the observed sites into five subregions based on their proximity (Table 3 and Fig. 9) and analyzed the density distribution in 0–1 and 1–2 m depths for each region (Fig. 10). In 0–1 m, densities in the NDF and DFNW regions and DFSE are normally distributed around averages of ~355 kg m$^{-3}$.
Layers with higher densities than the average (e.g., peaks around ~400 kg m$^{-3}$ at DFSE and the DFNW region) are wind-packed IHDFs, and layers with lower densities may originate from surface hoar or diamond dust (e.g., Koerner, 1971; Alley, 1988). In contrast, densities in the dome region and DK018 rarely exceed 400 kg m$^{-3}$ and are more often below 300 kg m$^{-3}$ in 0–1 m compared to the NDF, DFSE, and DFNW regions, resulting in the lower mean density. The shift of density distribution toward lower values may reflect snow deposition characteristics at the dome summit, which could be influenced by local
topography. For example, the absence of intense IHDFs (>400 kg m$^{-3}$) at the dome may be because wind-brown snow does not tend to ascend slope and reach the dome. In contrast, distinct ILDFs (<300 kg m$^{-3}$) may occur because surface hoar or diamond dust deposits predominantly under the calm condition of the dome summit. The ILDFs typically have fragile structures, possibly contributing to the density increase toward 2 m depth (Fig. 10b).




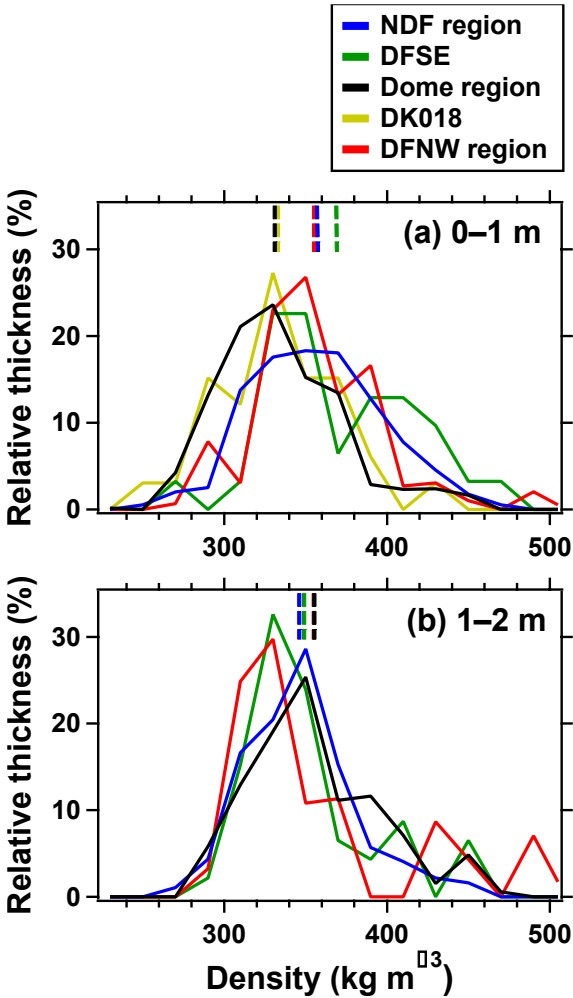

**Figure 10:** Density distribution of firn cores and pit walls in the five subregions around Dome Fuji. Solid lines indicate the relative cumulative thickness of cores and pit walls in each density range with a bin width of 20 kg m$^{-3}$ for (a) 0–1 m and (b) 1–2 m. The density data are derived from the firn cores ($\rho$) in this study and pit wall observations in literature: five profiles in the NDF region, one profile at DFSE, 15 profiles in the Dome region, one profile at DK018 (32 km northwest of the Dome Fuji Station), and three profiles in the DFNW region (see Table 3 for the site information and Fig. 9 for the density profiles). Dashed lines at the top of each panel indicate the average density for the regions.

## 4.2 Evolution of microstructural anisotropy

### 4.2.1 Effect of TGM on anisotropy

Microstructural anisotropy of ice and pores in the snow increases with TGM based on laboratory experiments (Pfeffer and Mrugala, 2002; Schneebeli and Sokratov, 2004; Srivastava et al., 2010; Calonne et al., 2014). The TGM may explain the depth





profiles of our $\Delta\varepsilon$ data in the Dome Fuji area, with the development of microstructural anisotropy within the top ~3 m (Fig.
4b), where seasonal and diurnal temperature variations create large vertical TGs up to ~15 K m$^{-1}$ (Fig. 4d). We can assume
that a fresh snow layer at the surface has homogeneous or horizontally elongated structure (e.g., Leinss et al., 2016) with $\Delta\varepsilon$
(axial ratio) of 0 (1), and $\Delta\varepsilon$ (axial ratio) increases to ~0.03–0.04 (~1.07–1.10) within the top few tens of centimeters, where
TG is the largest. Below ~3 m depth, where the TG is much smaller than the shallower depths, the vertical anisotropy decreases
with depth, probably because a part of the vertical alignment of grains collapses through grain packing (Fujita et al., 2009,
2014, 2016; Löwe et al., 2011).

The increase in the S.D. of $\Delta\varepsilon$ (Fig. 6b) and correlation coefficient between $\Delta\varepsilon$ and $\Delta\rho$ (or $\Delta$SSA) (Fig. 7a and 7c) within the
top ~3 m may be caused by selective developments of microstructural anisotropy in the IHDFs (Fig. 5 and 8). This observation
is consistent with earlier experimental evidence that the magnitude of TGM depends on the density of snow samples (Akitaya,
1974; Pfeffer and Mrugala, 2002; Schneebeli and Sokratov, 2004). For example, Pfeffer and Mrugala (2002) applied realistic
TG (20–80 K m$^{-1}$) to snow block samples with various densities (32–400 kg m$^{-3}$) to investigate the metamorphism with thin-
section stereological analysis and hardness measurement after three days. They found that the vertical bonds between snow
grains associated with snow hardening markedly developed in high-density snow during TGM. Thus, the density-dependent
TGM preferentially develops the microstructural anisotropy in IHDFs and may explain the correlation between the S.D.s in
the anisotropy and density below ~3 m depth (Fig. 6a and 6b).

The causes of the density dependence of TGM are unknown. We suggest that pore size, thermal conductivity, and air
permeability play key roles in the density dependence as follows. During the TGM, the smaller pore sizes in the IHDFs may
facilitate bond formation between the grains, while larger pores in the ILDFs are less likely to be filled with ice matrix (e.g.,
Akitaya, 1974; Marbouty, 1980). Furthermore, smaller grains in IHDFs should change their aspect ratios more easily by water
vapor condensation at the top or bottom of the grains. In addition, the higher thermal conductivity and lower permeability in
IHDFs (Calonne et al., 2011) result in lower TG and less efficient vapor transport through the IHDFs. This may result in net
transfer of mass from ILDFs to IHDFs and further facilitate vertical bond formation between grains in IHDFs, developing the
contrast of microstructural anisotropy between the layers.

### 4.2.2 Spatial relationship between accumulation rate and anisotropy

Our data show higher $\Delta\varepsilon$ at the southern sites (NDF and NDFN) than at the other sites (DFSE, DFS, and DFNW) (Fig. 4b).
Based on the discussion in Sect. 4.2.1, these microstructural features suggest that the near-surface firn at the southern sites
undergo enhanced TGM compared to the other sites. We discuss the effects of mean accumulation rate as possible cause of
the different magnitudes of TGM among the sites.





The accumulation rate determines the residence time of a firn layer in the top few meters where TG is large; thus, it should have an inverse correlation with the magnitude of TGM (Courville et al., 2007; Hörhold et al., 2009). The largest accumulation rate (DFNW) is 1.24 times higher than the lowest one (NDF). The average $\Delta\varepsilon$ for 0–0.5, 0.5–3, 3–6, and 6–9 m in the six cores are plotted against the accumulation rate for 1885–1992 C.E. (Oyabu et al., 2023) (Fig. 11a). Furthermore, we show the mean residence time of a layer in the top 3 m by dividing the water equivalent depth of 1050 mm, which is the average of the six cores at 3 m depth, by the accumulation rate (shown on the second bottom axis). $\Delta\varepsilon$ tends to be high at the NDF and NDFN sites for all the depth ranges compared to the other sites. In all the cores, the largest changes in $\Delta\varepsilon$ are observed between 0–0.5 and 0.5–3 m depths (black and red markers in Fig 11a), where the firn is exposed to large TG (Fig. 4d, also see Sect. 4.2.1). In addition, the magnitudes of $\Delta\varepsilon$ changes in the top 3 m appear to be negatively correlated with the accumulation rate; the increase in axial ratio converted from $\Delta\varepsilon$ is ~3 times larger at NDF and NDFN than at DFNW (Fig. 11a). The axial ratio for 3–6 and 6–9 m depths is larger at the lower accumulation sites than at the higher accumulation sites, suggesting that the spatial difference in the microstructural anisotropy developed by TGM are maintained in the deeper firn. Therefore, we conclude that the accumulation rate plays a significant role in controlling the magnitude of TGM around Dome Fuji, primarily through its effect on the residence time of a firn layer in the top few meters.

The sensitivity of $\Delta\varepsilon$ to accumulation rate is higher at lower accumulation rates (Fig. 11a), may implying the existence of positive feedback between TGM and microstructural anisotropy. The firn with more vertically elongated structure (created by TGM) become more permeable, thereby facilitating vertical vapor transport and potentially leading to stronger TGM (e.g., Albert, 2002). To confirm the non-linearity and this hypothesis, more data from sites with varying accumulation rates should be collected.



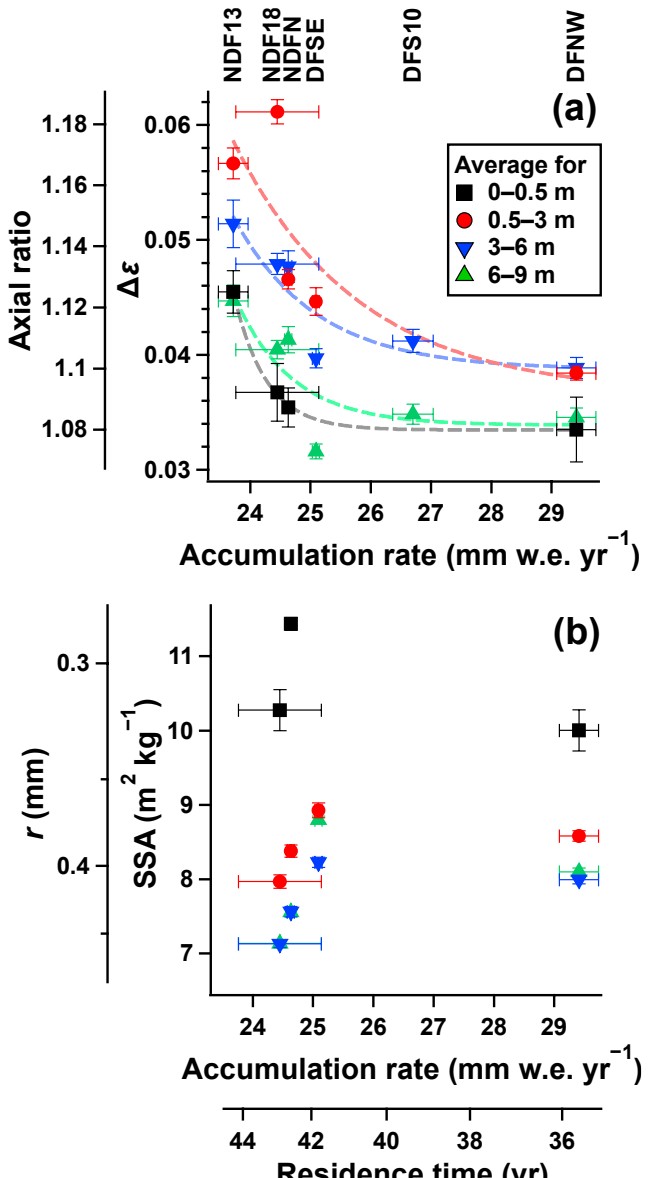

**Figure 11: Spatial relationship between Δε (or SSA) and accumulation rate. (a) Averages of Δε for 0–0.5, 0.5–3, 3–6, and 6–9 m of the six cores plotted against the accumulation rate. Vertical and horizontal error bars indicate standard errors of the Δε average and standard deviations of the accumulation rate, respectively (Oyabu et al., 2023). The Δε average is not calculated when there is a lack of data for more than one-third of each depth range. Dashed lines are exponential fitting lines for each depth range. (b) Same as (a) but for SSA. The second horizontal axis shows the residence time for which a certain layer stays in the top 3 m.**



### 4.2.3 Temporal relationship between accumulation rate and anisotropy

The 0.5 m moving averages of $\Delta\varepsilon$ for all the six sites show fluctuations on the scales from several tens of centimeters to meters (Fig. 4b). The fluctuations in microstructural anisotropy may reflect the variations of past accumulation rate, as observed in the Hercules Dome core (Hörhold et al., 2009). The age scales of our cores (Fig. 12a) are made by linear interpolation between the volcanic peaks (arrows in Fig. 12) with water equivalent depth (Oyabu et al., 2023). The six cores tend to show their maxima and minima in $\Delta\varepsilon$ around similar ages (maxima around 2005, 1975, 1940, and 1890 CE, and minima around 1990, 1955, and 1910 CE). The mean of the six cores (black line in Fig. 12a) also shows significant multidecadal fluctuations (exceeding 68% confidence interval for the six-core average based on the t-distribution). These results suggest spatial coherence in $\Delta\varepsilon$ on multidecadal timescales, which may reflect the temporal variations of mean accumulation rate in the Dome Fuji area.

We compare the fluctuations in $\Delta\varepsilon$ with past accumulation rates reconstructed for two sites by counting crust layers on pit walls assuming their formation in summers (Koerner, 1971; Hoshina et al., 2014) (Fig. 12c). The first site is the Plateau Station, located 216 km southwest of the Dome Fuji Station, covering 1842–1965 CE (Koerner, 1971). The second site is 3 km northeast of the Dome Fuji Station, covering 1958–2007 CE (Hoshina et al., 2014) (Fig. 1a). There is no significant difference in the mean accumulation rates between the two sites (the average and standard error for the Dome Fuji and Plateau locations for each period are $29.3 \pm 2.5$ and $27.8 \pm 0.1$ mm w.e. yr$^{-1}$, respectively). We use these data because of their high time resolution, although they should also contain short-term variability caused by local factors such as inhomogeneous deposition due to wind-driven redistribution and precipitation intermittency (e.g., Kameda et al., 2008). We calculated their 8-year moving averages, roughly corresponding to smoothing over ~0.5 m intervals (thick lines in Fig. 12c), to compare with the $\Delta\varepsilon$ data.

The 8-year moving averages of accumulation rates show maxima around 1990, 1955, and 1910 CE and minima around 2005, 1975, 1940, and 1890 CE (Fig. 12c). They are negatively correlated with the smoothed $\Delta\varepsilon$ (Fig. 12a). We test the significance of the negative correlation using a Monte Carlo approach. First, we detrended the smoothed $\Delta\varepsilon$ below ~1 m depth (called $\Delta\varepsilon_{\sim 1m}$) using the linear regression (dotted line in Fig. 12a). The $\Delta\varepsilon_{\sim 1m}$ and 8-year moving average of accumulation rates are randomly varied 1000 times according to their t-distributed probabilities, and linear regression is calculated for each pseudo data set (Fig. 13a). The mean slope is significantly negative ($-0.0009 \pm 0.0006$ / mm w.e. yr$^{-1}$). The negative correlation suggests that the microstructural anisotropy developed during low-accumulation periods (long exposure to TG), highlighting the importance of accumulation rates on the microstructural anisotropy.



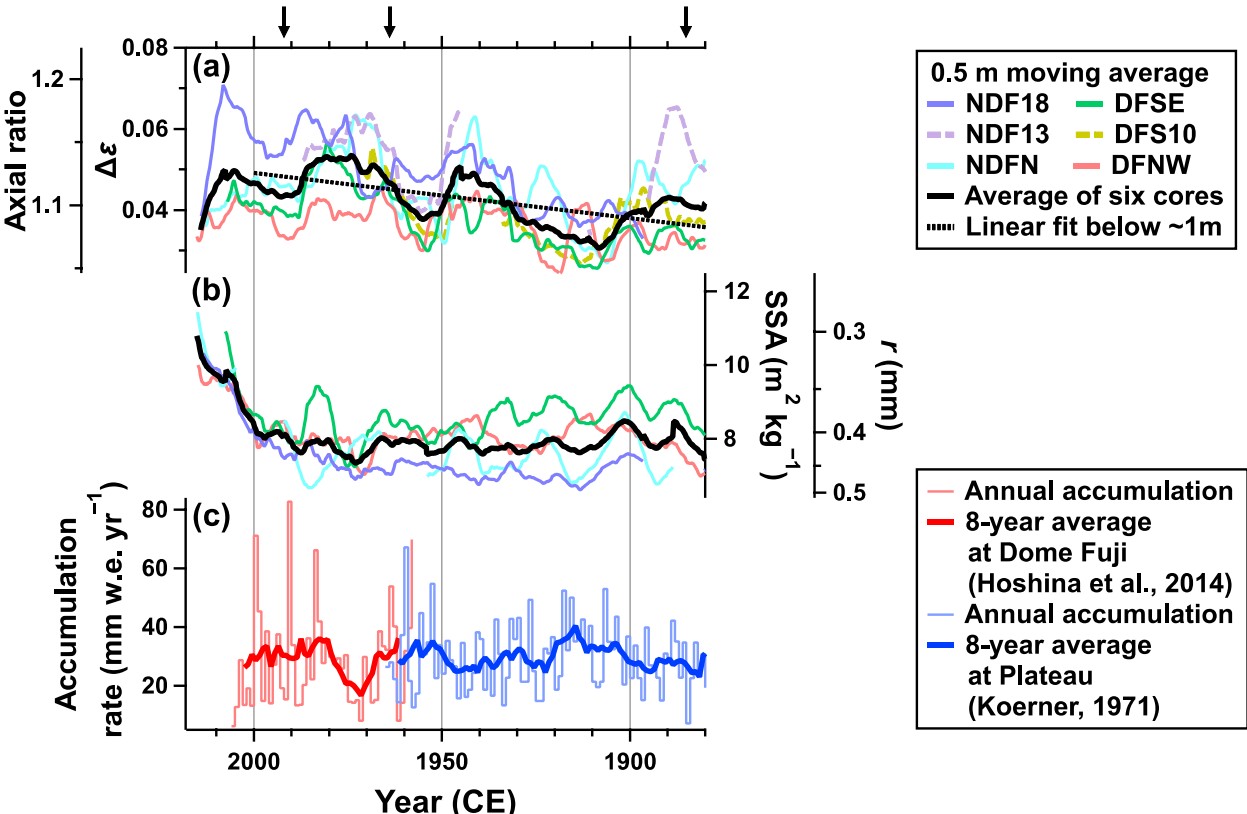

**Figure 12: Temporal change in Δε, SSA, and accumulation rate. (a) Moving averages of Δε for the six cores using a 0.5 m window (Same data in Fig. 4b but plotted against year (CE)). The age scales of the cores are made by linear interpolation between the volcanic peaks (black arrows) with water equivalent depths (Oyabu et al., 2023). The solid black line indicates the average of the six cores. The dotted line indicates a linear fit to the average before 2000 CE (corresponding to ~1 m). (b) Same as (a) but for the moving averages of SSA for the four cores (Same data in Fig. 4c). (c) The accumulation rates at Dome Fuji (3 km northeast of the Dome Fuji Station) (Hoshina et al., 2014) and the Plateau Station (Koerner, 1971), derived by counting summer crust layers on pit walls as annual layer boundaries. Thin and thick lines indicate annual accumulation and an 8-year moving average, respectively.**



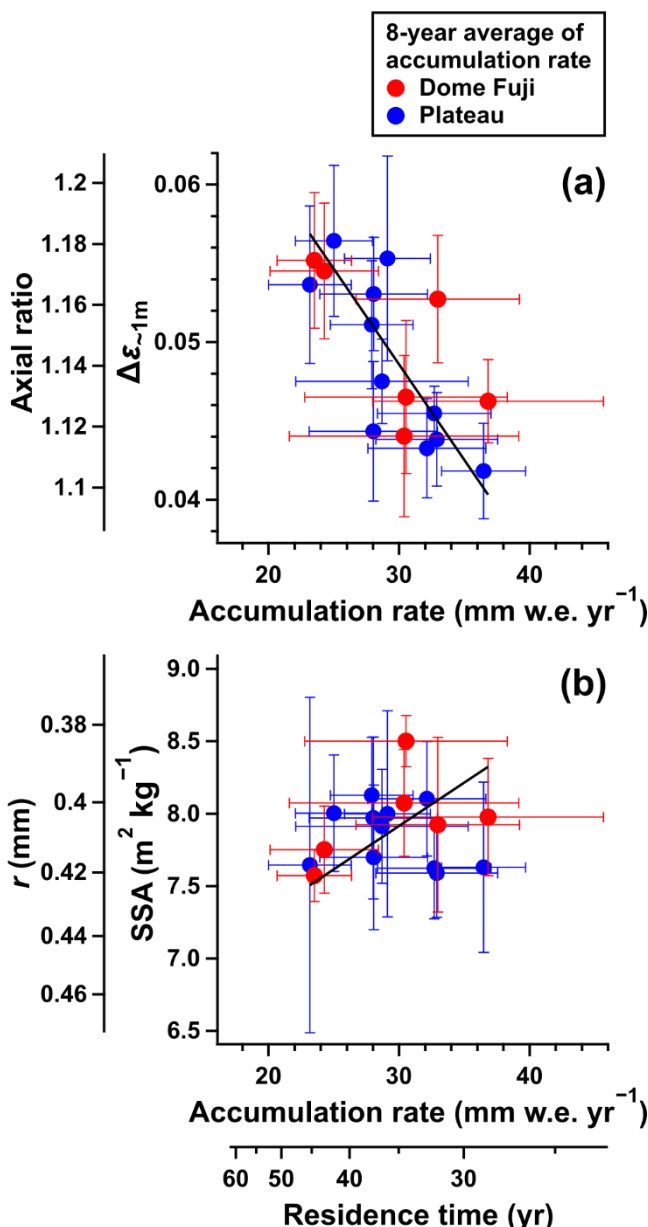

**Figure 13: Temporal relationship between Δε (or SSA) and accumulation rate. (a) Scatter plot of the Δε~1m (solid black line − dotted line in Fig. 12a) against the 8-year average of accumulation rate (thick line in Fig. 12c). (b) Scatter plot of the average SSA for the four cores (solid black line in Fig. 12b) before 2000 CE against the 8-year average of accumulation rate. Vertical and horizontal error bars indicate 68% confidence intervals based on the t-distribution for the Δε (SSA) average of the six (four) cores and the 8-year average of the accumulation rate, respectively. The black line is a standardized major axis regression to the data. The second horizontal axis shows the residence time for which a certain layer stays in the top 3 m.**





### 4.3 SSA evolution

#### 4.3.1 Effect of TGM on SSA

Grain growth (SSA decrease) during TGM has been observed in laboratory experiments (e.g., Marbouty, 1980; Calonne et al.,
2014) and the top few meters of firn at Antarctic inland (Gallet et al., 2011; Linow et al., 2012; Calonne et al., 2017). Our data
show decreases in SSA in the top ~3 m in the Dome Fuji area (Fig. 4c), consistent with the earlier studies. Below ~3 m depth,
the SSA decrease is not observed, probably because of small TG and the minor role of sintering for densification.

According to laboratory experiments, TGM is also expected to increase SSA variability (Akitaya, 1974; Marbouty, 1980;
Pfeffer and Mrugala, 2002; Schneebeli and Sokratov, 2004) (see Sect. 4.2.1). For example, Marbouty (1980) applied realistic
TG (25–66 K m$^{-1}$) to snow block samples with various densities (180–370 kg m$^{-3}$) and measured the grain size over a month;
less grain growth was observed for finer and denser snow. However, we do not observe a significant increase in the S.D. of
SSA in the top ~3 m of the four cores (Fig. 6c), indicating that the effect of TGM does not significantly change the initial
variability of SSA. The SSA is strongly correlated with density from the surface (Fig. 5, 7b, and 8), and both the S.D. of SSA
and density generally decrease with depth with similar large fluctuations (Fig. 6a and 6c). These results suggest that the SSA
is largely determined by the depositional environment at the surface, as is the case for density (see Sect. 4.1.2), and that wind
may play a key role. Strong winds may cause higher SSA and density by forming wind-packed layers composed of fragmented
fine grains (Domine et al., 2009).

#### 4.3.2 Relationship between accumulation rate and SSA

We discuss the differences in SSA among the studied sites and their possible causes. Our data show lower SSA at the southern
sites (NDF and NDFN) than the other sites (DFSE and DFNW) (Fig. 4c), possibly because of stronger TGM caused by the
longer residence time of a certain layer in the top several meters due to the lower accumulation rates, as discussed from $\Delta\varepsilon$
data (see Sect. 4.2.2). The average SSA for 0–0.5, 0.5–3, 3–6, and 6–9 m in the four cores are plotted against accumulation
rates for 1885–1992 CE (Fig. 11b). The largest changes in SSA occur between 0–0.5 and 0.5–3 m (black and red markers in
Fig 11b), where firn is exposed to large TG in the NDF18, NDFN, and DFNW cores (the DFSE core lacks the data for 0–0.5
m depths). Between the two depth ranges, SSA decreases more at lower accumulation sites; the increase in grain radius $r$
(second left axis in Fig. 11b) is ~2 times larger at NDF and NDFN than at DFNW. This stronger reduction in near-surface SSA
at the lower accumulation sites probably explain the lower SSA at NDF and NDFN than at DFNW in the deeper depths (3–6
m and 6–9 m). These results suggest that the SSA is indeed affected by accumulation rate through the residence time.

However, the correlations between SSA and accumulation rate for the four depth ranges are not as pronounced as those between
$\Delta\varepsilon$ and accumulation rate (especially for 0–0.5 m depths) (Fig. 11a), with SSA at DFSE showing the largest value despite





intermediate accumulation rate among the four sites (Fig. 11b). These results suggest that the post-depositional SSA changes
through the TGM do not fully replace the initial SSA variations determined by the depositional environments at the surface.
For example, the highest SSA at DFSE might be due to the lower surface topographic slope (Table 1), possibly leading to
preferential deposition of wind-driven segmented snow with fine grains (Domine et al., 2009). This is consistent with the
observation of more intensive wind-packed layers (higher $\rho$ and SSA layers) in the DFSE core compared to the other cores
(Fig. 3). To identify the environmental factors controlling the SSA at the surface and in the firn, more data from sites with
varying depositional environments (e.g., wind speed and topographic slope) are needed.

The spatial relationship between SSA and accumulation rate (albeit a weak relationship) might hold for their temporal
variations, as seen in the relationship between $\Delta\varepsilon$ and accumulation rate (see Sect. 4.2.3). However, the SSA data do not show
significant fluctuations on scales of several tens of centimeters or meters that covary in the four cores (Fig. 12b). Moreover,
no correlation is found between the mean SSA of the four cores and accumulation rate (Fig. 13b). The mean slope of linear
regressions for the 1000 times data set in a Monte Carlo approach (described in Sect. 4.2.3) is $0.01 \pm 0.04$ m$^2$ kg$^{-1}$/mm w.e.
yr$^{-1}$. Thus, we conclude that the role of TGM in SSA evolution in firn is minor compared to the depositional environmental
factors, which can significantly alter the initial SSA at the surface. This contrasts with the strong imprint of TGM on
microstructural anisotropy, which is essentially isotropic at the surface and develop primarily by TGM.


## 5 Conclusions

We continuously measured density, microstructural anisotropy, and SSA at high resolution (0.0025–0.02 m) for the top 10 m
of six firn cores collected within 60 km around Dome Fuji. These data provide the first detailed view on the evolution of
physical properties in a low-accumulation area (23–30 mm w.e. yr$^{-1}$). Furthermore, we investigated the spatial and temporal
variabilities in the physical properties of near-surface firn around Dome Fuji. Our main findings are summarized as follows:

(i) The density of the NDF18, NDFN, DFSE, and DFNW cores do not significantly increase within the top 4 m. The density
in the top 1 m is higher at the four sites (~355 kg m$^{-3}$) than near the flat dome summit or ridge (~330 kg m$^{-3}$), suggesting
that snow deposition patterns (e.g., wind-packing and frost) vary around the dome summit. The density variability (S.D.
of $\Delta\rho$) shows fluctuations with a decreasing trend over 10 m depth, with maxima appearing at similar depths in the four
cores. These maxima possibly formed during the periods with strong winds that can form wind-packed high-density layers.

(ii) The vertically elongated structure of the ice matrix and pores (i.e., microstructural anisotropy) is developed within the top
~3 m due to large vertical TGs caused by seasonal and diurnal temperature variations. The microstructural anisotropy is
more developed at the southern sites with lower accumulation rates than the northern sites with higher accumulation rates





around Dome Fuji. Furthermore, from the surface to ~10 m depth (roughly over the last century), the layers formed during the periods of relatively low accumulation may have developed higher microstructural anisotropy than those formed during the periods of relatively high accumulation. The negative correlations between microstructural anisotropy and accumulation rate suggest that the magnitude of early-stage, post-depositional microstructural development is primarily

controlled by the accumulation rate through TGM. The variabilities in microstructural anisotropy (S.D. of $\Delta(\Delta\varepsilon)$) increase within the top ~3 m, with the development of positive correlations between anisotropy and density, suggesting selective developments of microstructural anisotropy in IHDFs. The correlation between the variabilities (S.D.s) in microstructural anisotropy and density (or SSA) holds below ~3 m.

(iii) Grain growth, as indicated by SSA decrease, is observed in the top ~3 m probably due to TGM. Lower SSA (stronger TGM) is observed at the southern sites with lower accumulation rates around Dome Fuji. The spatial and temporal correlations between SSA and accumulation rate are weak, suggesting a minor role of TGM in the SSA variabilities along the cores. Instead, the initial SSA at the surface set by the depositional environments (such as wind) may be preserved and dominate the post-depositional SSA variabilities.


The density, microstructural anisotropy, and SSA in the top few meters of the studied sites may influence the densification rates in the deeper firn and the layering in the bubble close-off region to affect the gas fractionations of trapped air (e.g., Fujita et al., 2009). Understanding these processes in the deeper firn must be investigated in the future to better interpret gas signals in the deep ice cores.


**Appendix A: Conversion of NIR reflectance to SSA**

We constructed a regression curve between the NIR reflectance and SSA to convert the NIR reflectance measured by our optical line scanning system into SSA. To measure SSA, we used the Handheld Integrating Sphere Snow Grain Sizer (HISSGraS) (Aoki et al., 2022), which has the same measurement principle as the IceCube widely used to measure snow SSA.

(Gallet et al., 2009) but can directly measure the plane surface of snow samples. It measures hemispherical NIR reflectance and converts it into SSA using a radiative transfer model. The spatial resolution of the measurement is ~0.025 m. Here, we measured the SSA at 0.02 m intervals and NIR reflectance using our system on 19 core samples from 0–60 m depth of the NDFN core. Figure A1a shows the result at 4.21–4.41 m depth as an example. We confirmed that the NIR reflectance correlates with SSA with both showing higher values at shallower depths. Figure A1b shows a quadratic regression curve derived from

all the NIR reflectance and SSA data:

$$SSA = 0.0131 \, NIR^2 - 1.20 \, NIR + 30.1 \tag{A1}$$





The root mean square error of the regression curve is 1.0 $m^2$ $kg^{-1}$. The error of SSA measurement is ~15% for 6–15 $m^2$ $kg^{-1}$ corresponding to the SSA in the top 10 m of the measured four cores.

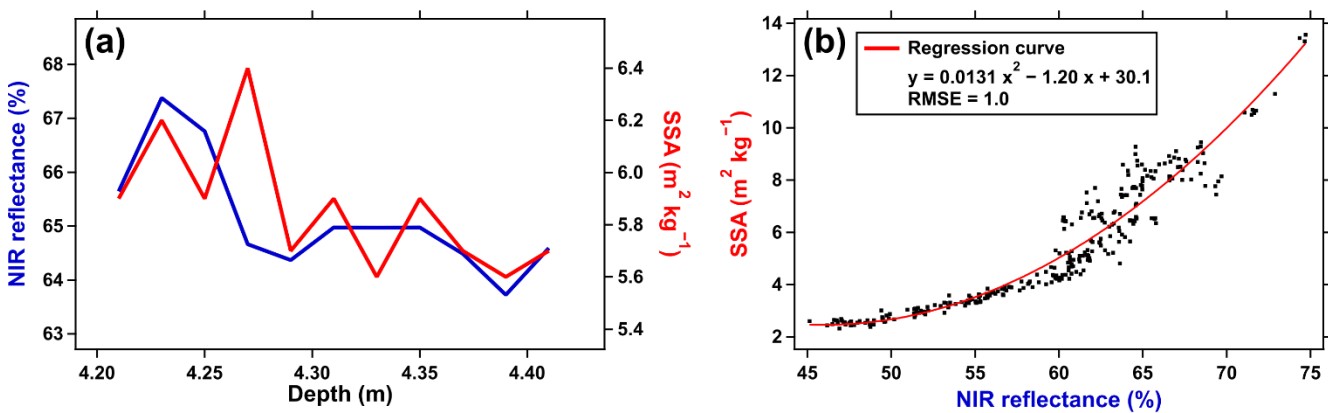

**Figure A1: (a) NIR reflectance measured using our optical line scanner system (blue) and SSA measured using HISSGraS (red) at 4.21–4.41 m of the NDFN core. Data at 0.02 m intervals are shown. (b) Relationship between NIR reflectance and SSA. The red line indicates a quadratic regression curve.**



### Data availability

All data presented in this study are available at the NIPR ADS data repository (https://ads.nipr.ac.jp/dataset/A20230807-001 during revision).

### Author contribution

RI and SF designed the laboratory experiments and carried them out (RI measured the NDF, NDFN, DFSE, and DFNW cores. SF measured the NDF13 core). RI processed and analyzed the data, and wrote the manuscript with inputs from other authors and editing by SF and KK. SF developed the measurement methods. KK, IO, NK, and HM collected the firn core samples. All authors contributed to the discussion and reviewed the manuscript.

### Competing interests

The authors declare that they have no conflict of interest.



## Acknowledgments

Field campaigns were conducted as part of the Japanese Antarctic Research Expeditions (JARE), supported by the National Institute of Polar Research (NIPR) under the Ministry of Education, Culture, Sports, Science and Technology (MEXT). We thank all members in the fieldworks who contributed to obtaining the firn cores, field logistics, and processing. We thank
Kyohei Yamada for on-site support during the NDFN core drilling, Teruo Aoki (NIPR) for constructive comments about SSA measurement, and Neige Calonne (WSL) for providing the 3 m pit data at Point Barnola.

## Financial support

This study was supported by the Japan Society for the Promotion of Science (JSPS) and Ministry of Education, Culture, Sports, Science and Technology (MEXT) KAKENHI (grant no. 18H05294 to Shuji Fujita and 17H06320 to Kenji Kawamura) and
JST FOREST Program (grant JPMJFR216X to Ikumi Oyabu).

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
