# Peer review of "Spatial distribution of vertical density and microstructure profiles in near-surface firn around Dome Fuji, Antarctica"

_EGUsphere, 2023_

## Author Comment (AC1)

The authors would like to thank Z.R. Courville for recognizing the value of our study and providing valuable comments and feedback on our manuscript. We carefully revised the manuscript and engaged in correspondence for all of your comments. Our responses are provided in blue. The sentences in red are the revised text.

5  Reviewer #2: Z.R. Courville

Overall, this is a very interesting and impressive set of high resolution firn physical property measurements from relatively closely spaced (i.e., within 60 km of one another) low accumulation rate sites in Eastern Antarctica. The paper presents a suite of complementary measurements of high vertical resolution density, anisotropy and SSA. Some of the more interesting conclusions of the paper include that there may exist a
10  positive feedback between temperature gradient metamorphism and anisotropy, which makes sense intuitively and is supported by the results, and that post-depositional changes in SSA are not overcome by initial SSA variations formed by surface depositional conditions. The paper also presents evidence that at least for this low accumulation site that there is a lack of significant density in the top 4m of the firn column and that firn at the summit of the dome is less than the surrounding area, and points to the role of wind and,
15  related to wind, topography, in the determination of initially high density layers which impact firn densification rates in addition to snow accumulation and temperature (as are typically used to determine firn density with depth profiles empirically). Furthermore, the paper does an excellent job of describing the environmental factors driving the density and microstructure variations in the firn in the context of previous studies in order to interpret the results, and discuss the interrelation of the snow microstructural parameters
20  (i.e., pore and grain size and anisotropy) and environmental conditions to explain density variations.

**Science and methodology questions:**

Table 1: How well does the NDF18 10 meter temperature reflect the 10 m temperatures at NDF13 and NDFN? Same question for the Dome Fuji temperature being used for the DFS10 site (and likely more
25  differences between the actual values for these two sites)?
We agree with your point that 10 m temperatures at the NDF13, NDFN, and DFS can be different from neighboring sites. We will remove them from the table 1.

Line 165: How were the cores transported? Specifically, what temperatures were the cores shipped and stored at and what measures were taken to ensure minimal grain size changes?
30  We will add the following sentences: "*The cores were transported within a temperature range of −30°C to −15°C during the transport from the field to a ship (for ~20 days, of which ~5 days above −20°C) and at −28°C during the ship transport (for ~80 days), then stored at −30°C.*" around L140 (in the original text).
     We did not apply specific measures to minimize the grain growth during transportation. Instead, we will consider the effect of grain growth as follows. "*The SSA decrease due to metamorphism during sample*
35  *transportation is expected to be less than 15% if the SSA of snow is less than 15 m² kg⁻¹, according to the*

*empirical SSA reduction rate (Taillandier et al., 2007). Although metamorphism during transportation may cause a systematic error, it does not affect our discussions on relative variability in SSA (e.g., differences among sites and variations within each core).*" in L244.

Figure 8: not sure if this figure is needed? It is confusing to interpret, and seems to be conveying much of the same info that is in Figure 7.

Figure 7 shows the evolution of relationship between density, $\Delta\varepsilon$, and SSA in the ILDF and IHDF using a statistical indicator. In the $\rho$-SSA scatter plots in Fig. 8, we can directly observe the evolution of the three variables and their relationship on scatter plots, helping us understand that ILDF and IHDF evolve with different initial conditions and different evolution processes. Thus, we believe that Fig. 8 is valuable for presentation in the paper. Nevertheless, we agree that interpreting this graph might be confusing in the flow of the main text, thus we will move Fig. 8 to Appendix.

**Minor/technical issues:**

Line 30: "For example, accurate density profile of near-surface firn is essential to derive the surface mass balance from the change in surface height" should be "For example, an accurate density profile of near-surface firn is essential to derive the surface mass balance from the change in surface height.." (just missing the article "an")

We will correct it.

Figure 1: This is a really great figure that really helps to clarify the context of wind and accumulation.

Thank you for recognizing the effectiveness of the graph.

Table 1, caption for "a" should be "The number after the letter designation…" or "The number in the alphanumeric designation.."

We will correct it. →"*The number after the letter designation…*"

Line 185 (Figure 2 caption): should be, or more precisely, is more commonly written "perpendicular to the page"

We will correct it.

Line 538: "suggesting that the spatial difference in the microstructural anisotropy developed by TGM are maintained in the deeper firn." Should be, "suggesting that the spatial differences in the microstructural anisotropy developed by TGM are maintained in the deeper firn."

We will correct it. →"*differences*"

Line 541: "The sensitivity of $\Delta\varepsilon$ to accumulation rate is higher at lower accumulation rates (Fig. 11a), may implying the existence of positive feedback between TGM and microstructural anisotropy." Should be "The sensitivity of $\Delta\varepsilon$ to accumulation rate is higher at lower accumulation rates (Fig. 11a), may imply the existence of positive feedback between TGM and microstructural anisotropy."

70    We will modify it to "*The sensitivity of Δε to accumulation rate is higher at lower accumulation rates (Fig. 11a), implying the existence of positive feedback between TGM and microstructural anisotropy.*"

Line 542: "The firn with more vertically elongated structure (created by TGM) become more permeable, thereby facilitating vertical vapor transport and potentially leading to stronger TGM (e.g., Albert, 2002)."
Should be, "The firn with more vertically elongated structure (created by TGM) becomes more permeable,
75    thereby facilitating vertical vapor transport and potentially leading to stronger TGM (e.g., Albert, 2002)."
We will correct it. →"*becomes*"

Line 724: fieldworks should be fieldwork
We will correct it.

---

## Author Comment (AC2)

The authors would like to thank Ghislain Picard for recognizing the value of our study and providing valuable comments and feedback on our manuscript. We carefully revised the manuscript and engaged in correspondence for all of your comments. Our responses are provided in blue. The sentences in red are the revised text.

Reviewer #1: Ghislain Picard

Review of "Evolution of layered density and microstructure in near-surface firn around Dome Fuji, Antarctica" by Ryo Inoue et al.

The article presents and analyzes measurements of the snow microstructure in the top 10 m of the firn around Dome Fuji in Antarctica. In total, 6 cores retrieved within 30 km are analyzed with unique carefully-designed techniques at 2 cm resolution for three important properties: density, specific surface area, and dielectric anisotropy. The analysis aims to explain the vertical profile characteristics and the spatial differences as a function of the meteorological variations (accumulation mainly). This study sheds light on post-depositional processes and snow metamorphism operating in the dome regions in East Antarctica and provide experimental evidences from the surface to help understand firnification processes operating at depth. While the results mainly confirm previous work, the empirical material is abundant, hence allowing a very detailed analysis that draws almost everywhere interesting and stronger conclusions. This study is important because the processes dominating the snow evolution in this region are very different and poorly investigated compared to other regions in the world (alpines and arctic regions). The role of the post-deposional processes / bloxing snow is highlighted. The targeted audience is polar snow and ice core scientists. It is in the scope of The Cryosphere.

The paper is well written and the structure is logical. The figures and tables are abundant and adequate. Only changes are required to improve the readability of the challenging Fig 3 that presents all the raw available materials. The analysis is overall carefully conducted with some statistical information. However, I've noted many minor issues, that result in a large number of comments below that may require significant work. Despite this, I'd like to stress that the overall quality and the value of all these rare observations for the snow and ice core community are very high. This is why I strongly recommend the publication of this study and of the open dataset.

Detailed comments:

        Regarding the title:

        - "Evolution" evokes time to me rather than the vertical spatial dimension.
        We appreciate your valuable feedback and will change the title (see below). We will replace "evolution"
        with "vertical profiles" in L34, L121, L390, and L658 in the original text.

- The horizontal spatial component of the study (the 6 cores in a >30km region) is not present while I
        think this is a value of this study.
        We will change the title.

        - "layered". As it is apparent in my comments below, I believe (this is an opinion) that the snowpack on
        the High Plateau is not layered, due to the tremendous and permanent redistribution by wind. This paper
draw conclusions in this direction as well to my understanding. The use of "layer" should therefore be
        reduced as much as possible because it conveys a message that reinforces the unconscious and dominant
        picture of the highly layered alpine /arctic snowpack that most of snow scientists know by daily
        experience, far from the complexity of the high plateau in Antarctica that only a few expeditioners have
        experienced. I let the authors agree or not on this point, and make the necessary changes if appropriate. It
is a subjective comment.
        The perception of the term "layering" can be different. As you mentioned, some scientists interpret
        "layering" as snow that spreads widely in the horizontal direction and retains seasonal variations.
        However, in inland plateau regions, due to wind redistribution, there is less horizontal extent, and snow
        with different physical and chemical properties accumulates in patches. Scientists focusing on the plateau
regions commonly call the patchy snow a single "layer" (e.g., Hörhold et al., 2011; Calonne et al., 2017).
        To maintain consistency with previous studies, we use "layer" in this sense.
                However, to avoid the misunderstanding that you mentioned, we will remove the word "layered"
        from the title, and change the word "layer" to other words in the main text in L119. We will also explain
        the word "layering" in the context of this study where it is first introduced in the manuscript (please see
our reply to comment L46).

        Overall, I suggest to include the idea of "vertical profiles" and "spatial distribution" in the title.
        We will modify the title to "*Spatial distribution of vertical density and microstructure profiles in near-
        surface firn around Dome Fuji, Antarctica*"

Detailed comments:

L46: "the layers are tens of centimeters thick". To my experience of the Dome C area, especially sampling a 50m long trench, it is even difficult to identify layers. Considering the wide audience of The Cryosphere, and the widespread belief that snow accumulates as layer everywhere, I'd suggest the authors (if they share the same experience and point of view) to elaborate this point that may lead to incorrect assumptions in modeling. For instance Picard et al. 2019 showed that the age of snow on the surface can vary spatially on short scales from 0 to 300 days, with potential consequences on the microstructure, isotopic composition, etc of that snows.

Thank you for the reference. We acknowledge that snow accumulates in patches also at Dome Fuji. To avoid incorrect assumptions in modeling, we will expand the explanation on the heterogeneous deposition nature in the following text:

"*In contrast, in low-accumulation areas (< ~50 mm w.e. yr$^{-1}$, represented by dome areas on the East Antarctic Plateau), the layers are tens of centimeters thick with several meters of horizontal extent, and do not show seasonal cycles (e.g., Hörhold et al., 2012; Picard et al., 2019). For example, the age of surface snow can vary from 0 to 300 days within less than 100 m of horizontal distance at Dome C, leading to different initial physical properties of the snow layers (Picard et al., 2019). Also, surface mass balance can vary from −30 to 100 mm w.e. yr$^{-1}$ within 100 m of horizontal distance at Dome Fuji (Kameda et al., 2008). These facts indicate that snow redistribution, erosion, and precipitation intermittency disrupt the homogeneous seasonal snow deposition; instead snow deposits in patches in low-accumulation areas.*"

L49. "the firn layers undergo metamorphism over time by packing and rounding of snow grains". While this sentence is true in general, and applies well to alpine snow, on the Antarctic plateau rounding is very rare at depth to my experience, owing to the permanent temperature gradients >20 K/m.

We acknowledge that temperature gradient metamorphism plays a crucial role in firn metamorphism at depth, as our study also concluded. On the other hand, we believe that equi-temperature metamorphism is also a fundamental process particularly for fresh snow at the surface on the Antarctic plateau and is worth mentioning. In the revision, because "rounding" may be misleading for buried snow, we will replace the phrase "*by packing and rounding of snow grains*" with "*by vapor diffusion to decrease the surface free energy*".

L62 Mord → Maud
We will correct it.

L82. "in the Megadune region" → "a Megadune region". This region is not unique, megadunes are widespread in inner Antarctica.
We will correct it. → "*a megadune region*"

L94-97. "detailed observations of firn microstructure" is subjective, may be I do not understand the idea expressed by the authors, but they are many measurements of SSA and density (i.e. beyond visual inspection) published in (self citation): Picard et al. 2014, Libois et al. 2014, Picard et al. 2022, Leduc-Leballeur et al. 2015, Brucker et al. 2009. The SumUP database also contains density measurements.

Thank you for informing us of several related studies in the region with a longitude 100–140°. We will modify the text to the following: "*High-resolution microstructure profile in the top few meters has not been investigated beyond visual inspection in the Dome Fuji area.*"

The studies will be cited in L88: "*The SSA measurement using near-infrared light has also been applied to polar firn (e.g., Gallet et al., 2011; Libois et al., 2014; Picard et al., 2014, 2022).*"

L110. I suggest to add a reference here for the gamma ray technique.

We will add references "*(e.g., Gerland et al., 1999; Hori et al., 1999)*"

L120. The abstract is mentioning "six cores" which may be misunderstood with "five sites", so maybe add "six cores" here as well.

We will modify the part to "*properties among six firn cores collected from the five sites*"

147. Using ERA5 to investigate spatial variability with 30 km is inadequate, due to the coarse resolution of this dataset and the even coarser resolution of the underlying models. I suggest to remove the information coming from ERA5. It is likely that the conditions are more variable than registered in ERA5, especially for the wind speed.

We agree with your suggestion and will remove the information from ERA5 and related explanations in the main text (L144–148).

L165. What about the conditions of transport (temperature?). The SSA can change quite within month at temperature higher than ~-35°C.

We will add the following sentences: "*The cores were transported within a temperature range of −30°C to −15°C during the transport from the field to a ship (for ~20 days, of which ~5 days above −20°C) and at −28°C during the ship transport (for ~80 days), then stored at −30°C.*" around L140 and "*The SSA decrease due to metamorphism during sample transportation is expected to be less than 15% if the SSA is less than 15 $m^2\,kg^{-1}$, according to the empirical SSA reduction rate (Taillandier et al., 2007). Although metamorphism during transportation may cause a systematic error, it does not affect our discussions on relative variability in SSA (e.g., differences among sites and variations within each core).*" in L244.

L178. How the error is determined ? May be add a reference.

We will add references "*(e.g., Gerland et al., 1999; Hori et al., 1999)*". The densimeter we used employs Beer's law, which is also employed by the previous studies. It determines the sample density using the sample thickness and the intensity of penetrated gamma-ray through the sample. We determined the error of the density by the root mean square of relative errors in the variability of sample thickness (1%) and the variability of gamma-ray intensity during a typical firn measurement (1.5%).

L198. Given the non symmetrical geometry in the resonator and of the sample, I'd expect that the measurements in the two axis are subject to different error (bias and random variations). How are the vertical and horizontal measurements intercalibrated ?
Regarding random errors, the S.D. of permittivity at eight different frequencies within 15–20GHz for both the vertical and horizontal components are the same (0.003).

Regarding systematic errors, our resonator is designed to be perfectly symmetrical in geometry and curvature, thus unlikely to cause different errors for the vertical and horizontal components.

       Regarding sample asymmetry, the Gaussian beam at 15–20GHz passing through a sample has a diameter of ~0.038 m (theoretical value), at which the beam intensity (light energy that passes through a specific area in unit time) becomes $1/e^2$ of the peak intensity at the center of the beam. The narrowest part of our sample width is ~0.060 m (Fig. 2a in the manuscript), so the intensity of the beam hitting outside of the sample is only 0.7% of peak intensity.

       Additionally, we tested the error resulting from beam hitting outside the sample by narrowing the width of a firn sample (0.09 m wide and 0.03 m thick) by a few millimeters and measuring the permittivity each time. We found that when the sample width becomes 0.06 m, the measured permittivity decreases by up to 0.0025 (< S.D. of eight measurements, 0.003) compared to the sample with 0.09 m wide. Since most part of our sample width exceeds 0.06 m (Fig. 2a), the systematic error in the horizontal component of permittivity is expected to be well below the random error.

       Given the above considerations, we currently do not calibrate either the vertical or horizontal component. Further investigation into each error factor would be future work.

L210. I'd expect that some theoretical mixing formula could be used here instead of empirical fits. The comparison below using Polder van Santen mixing formula and Mätzler 2006 ice permittivity formula as a function of temperature shows the agreement of Oyabu's -16°C curve, but not -30°C curve. The temperature dependence seems very important for such a small difference in °C.   This potentially leads to density estimation differences larger than the error stated in L241. 6 – 14 kg m-3. I suggest   the authors to re-analyse the derivation of the relationship at -30°C and its experimental error.
We also plotted the relationship between permittivity and density with different formulas in the right figure, and found that Oyabu's −30°C curve agrees with the PvS curves (perhaps some error in the reviewer's calculation). We also note that Oyabu's calibration curves are empirically derived using the same firn samples as used in this study, thus they take into account any bias due to instrument and grain geometries, and may be

somewhat different from the previously published curves and theoretical ones. We believe that Oyabu's empirical formula is most suitable for this study, but it may not be universally applicable to other instruments and samples.

L240. In this optical configuration (directional-directional reflectance), it is likely that the reflectance depends on density in addition to SSA, especially for low densities. Given that Fig A1 shows a very significant variability for high SSA (correlated with highly variable and low density in the cores), I suggest to explore this dependency, which is possible with this large dataset.

Dot colors in the right figure represent density. For a certain SSA range, variability in reflectance is not dependent on density; e.g., data with SSA ranging from 8 to 10 $m^2$ $kg^{-1}$ (box in figure below) shows no significant correlation between reflectance and density ($r = 0.15$) (the variability may be due to sample quality or some other unknown factors). On the other hand, the reflectance with SSAs larger than 6 $m^2$ $kg^{-1}$ might be biased toward low values due to low density (<400 kg $m^{-3}$), of which sparse structure allows near-infrared light to pass through the sample. However, this systematic error does not affect our discussions on relative variability in SSA (e.g., differences among sites and variations within each core), and thus we currently do not apply any calibration for density dependence. Further investigation of the density dependency in our method would be future work.

[Figure]

L240. Adding a figure comparing the hemispherical reflectance and the directional reflectance of the 60 samples is also useful in order to distinguish (and quantify) the error due to the non-hemispherical reflectance and that due to the inherent imprecision of the SSA – hemispherical reflectance relationship.

We will add curves of the relationship between SSA and hemispherical (or directional) reflectance derived from a radiative transfer model (Aoki et al., 1999) in Fig. A1b (graph below). The reflectance of samples measured by our system mostly fall within the two curves, suggesting that the reflectance has intermediate property between the two reflectance. Since the reflectance measured by our system is empirically converted to SSA (red line), the fact that the reflectance is non-hemispherical is not a source of error.

Regarding SSA measurement with HISSGraS that also measures NIR reflectance, the measured reflectance comprises directional and hemispherical reflectance and is converted to SSA using SSA–reflectance relationship that takes into account the fraction of the two reflectance components (Aoki et al., 2023). Thus, in theory, no errors arise from the actual reflectance being different from the hemispherical reflectance. The term "*hemispherical*" in the phrase "*between the reflectance measured by*

*our system and the SSA obtained from hemispherical NIR reflectance*" in L240 was miswritten (we are sorry for any confusion caused by this miswriting) and modified to "*between the reflectance measured by our system and the SSA measured using the Handheld Integrating Sphere Snow Grain Sizer (Aoki et al., 2023)*".

[Figure]

L240. In addition, because of the two apparent distinct regimes in Figure A1, the main text should be amended to provide separate error values for SSA< 5 m2 / kg and SSA > 5 m2/kg. In fact, the 15% error is an average but does not highlight this major difference of regime. Also you may indicate the range of applicability of the fit in the main text (SSA < 14   m2/kg). It is not uncommon to get higher SSA near the surface in the ridge region.

According to the suggestion, we will revise L242 as follows: "*The systematic error in SSA measurement is $\pm 2\ m^2\ kg^{-1}$ for SSA less than $20\ m^2\ kg^{-1}$ (Aoki et al., 2023), and the error for the regression curve for calibration is $0.4\ m^2\ kg^{-1}$ for SSA between 2 and $5\ m^2\ kg^{-1}$ and $0.9\ m^2\ kg^{-1}$ for SSA between 5 and $14\ m^2\ kg^{-1}$ (Appendix A).*" and will revise L240 as follows: "*Subsequently, we converted the reflectance into SSA using an empirical relationship between the reflectance measured by our system and the SSA measured using the Handheld Integrating Sphere Snow Grain Sizer (Aoki et al., 2023), which is applicable for SSA less than $14\ m^2\ kg^{-1}$*"

L275. I find the notation kg m-3 m-1   clearer than with the power of 4. A matter of preference.
We will modify it in L275, L276, and Table 3.

L277. "agree with each other within the measurement error". It seems that the variability is larger than the error, isn't it ? Maybe add the error as shaded areas in the Fig 4a plot, at least in your response to the reviews, because I acknowledge it might be too difficult to read for the main text.
In the figure below, we added shade representing an error of 9 kg m$^{-3}$ (*we corrected the error to the value in the top 10 m* estimated by Oyabu et al. (2023)). As you pointed out, the differences between the cores at the same depths exceed the error estimate in some cases (e.g., green curve at 2.5m, and light blue curve at 3.5m). Thus, we will modify this part to "*show no systematic differences exceeding the measurement error.*"

[Figure]

L281   "on a scale of ~0.1 m, reflecting the density layering of firn (Fig. 3)". It is not visible in Fig 3 that the variation scale is 0.1 m, the individual measurements are not represented. A more precise quantification of the correlation length, using an AR(1) model for instance, would be valubale as the layering (and its origin) are crucial in ice core interpretation (interpretation of isotops f.i.).

The correlation length for the six cores ranged from 0.04 to 0.06 m, which seems too low compared to our visual inspection. The lack of longer correlation length may be due to the lack of regular intervals between the layers (e.g., annual layers). Thus, we feel that the correlation length does not represent the major (quasi-) periodicity of significant density variation and may lead to potential misunderstanding on the origin of the layers by the readers.

We also calculated the mean period of density fluctuation in the NDF18 core by manually counting the number of maxima (vertical lines in the figure below). The distribution of the periods is also shown in the figure below. The mean period is 0.11 ± 0.04 m (mean and S.D.). Taking into account that the tail of the distribution extends to the period of ~0.3 m, we will modify the part to "*on the scales of ~0.05–0.3 m*".

[Figure]

[Figure]

L281-290. To interpret the S.D., the error on the S.D estimator should be calculated. Given the large amplitude of the S.D. and the relatively small number of observations (50 per meter at,most) it is possible that the S.D estimates are subject to uncertainties of the order of the interpreted differences. In fact, a question is whether the oscillations of the S.D. in Fig 6 are real or statistical artifacts.

The 95% confidence interval of the S.D. of density is ±5 kg $m^{-3}$ based on the chi-square distribution (shade in the figures on the right). This is significantly smaller than the variations of S.D. (typically >10 kg $m^{-3}$), thus we think the oscillations of S.D. in Fig 6 are real. We will add the statistical information in L287: "*In addition, the moving S.D. fluctuates by ~10–20 kg $m^{-3}$, which is larger than the 95% confidence*

*interval of ±5 kg $m^{-3}$ based on the chi-square distribution*".

[Figure]

L337. Similar question. Are these oscillations real or due to a few particularly high variations in some specific layers. Based on Fig 5, I see a few infrequent large variations. Are they the causes of these oscillations ?

Specific layers with high $\Delta\varepsilon$ increase S.D., and such layers tend to concentrate in a 1 m window at a certain depth and vice versa; e.g., fluctuations with amplitudes larger than 0.02 occur several times in 5–6 m, but do not appear at all in 6–7 m for the DFNW core (Fig. 5d), which are the major cause of the oscillations in S.D. We think the oscillations are meaningful, reflecting the past environment that led to the frequent formation of layers with high $\Delta\varepsilon$ (related to our response to comment L454).

L345. Since Epsilon is related to density, it is expected that part of Delta Epsilon changes with density. That is, when the density increases both vertical and horizontal permittivities logically increase, and since the horizontal one is typically larger than the vertical one, the difference should increase as well. To separate this "obvious" contribution from the anisotropy contribution, which is the one of geophysical interest, is it possible to investigate the ratio Delta Epsilon / Epsilon instead of the difference ? This comment is also related to the discussion L 505-510.

We plotted the "the ratio Delta Epsilon / Epsilon" of the NDF18 core in the figure below. The decreasing trend down to 10 m and peak clarity of $\Delta\varepsilon / \varepsilon_h$ are almost the same as those of $\Delta\varepsilon$, suggesting that the effect of density on $\Delta\varepsilon$ is quite small. Using either $\Delta\varepsilon / \varepsilon_h$ or $\Delta\varepsilon$ does not affect our conclusion, so we use $\Delta\varepsilon$ following Fujita et al., 2009; 2014; 2016.

[Figure]

L359. I suggest to change "at the surface" by "near the surface" because it is unlikely the surface SSA is so low (11 m2 /kg). Satellite optical observations show much larger SSA values in this area of the Antarctic. Also, to my experience in the field, SSA measured on the surface and along snow core very close to the surface always shows a large difference. The conditions of transport from Antarctica to Japan may also have altered the highest SSA.

We will modify the term accordingly.

L396-397. "Generally, the density of near-surface firn is expected to increase with depth due to settling of snow grains under overburden pressure". Overburden pressure is not the main process of densification near the surface to my knowledge. At least in Alpine snowpack most of the early stage densification is due to metamorphism (packing caused by self-gravity of the grains), without requiring any pressure, which is insufficient anyway near the surface. The observations of the paper conform with the common knowledge, the first sentence should be revised to remove the opposition with the second sentence.

We agree with the comment and will remove the sentence "*Generally, the density ... . However,*". We also think that it is worth emphasizing the lack of density increase in the top 4 m despite ~50–65 years after deposition (regardless of the densification mechanism); thus, we modify the second sentence (first sentence in the revision) to "*Our ρ (or ρ_ε) data in the NDF18, NDFN, DFSE, and DFNW cores do not show significant increase in the ~0–4 m range, despite 50–65 years after deposition (Fig. 4a)*."

L406. "However, slow densification due to the slight overburden pressure cannot explain the observed density decrease for the top ~2 m".   I don't understand "however". How does this oppose with the previous sentence ?

*However*" modifies "*decrease*", which is the opposite effect of densification. To make this clear, we will modify it to "However, the low densification rate cannot explain the *decrease* in density for the top ~2 m"

L414. Champollion et al. 2019 investigates decannal temporal variations at Dome C, there are indeed surprisingly large, but I'm not sure if they can explain the data in Fig 3a. Note that the same methodology could be applied to Dome F, but this is certainly out of the scope of the present paper and only provide a few decades (a few meters). There are certainly other references to illustrate the same point, I'm sorry for the high number of self citation in this review.

We are unsure whether the ~100 kg m$^{-3}$ decrease in density for the top 3 cm from 2002 to 2011 (Champollion et al. 2019) can explain our density profile (maybe inconsistently forming an increasing density profile with depth). On the contrary, pit density observations in 1968, 1994, and 2007 CE around Dome Fuji (~5 pits for each campaign) have shown a consistent mean near-surface density of ~350 kg m$^{-3}$ (in the top 1 m). Thus, it might also be possible that the variability of surface density around Dome

Fuji has been rather small over the past half-century. Considering these points, we will revise L414 as follows:

"*Another possibility may be that the lack of mean density increase in the top few meters reflects the increasing trend of the surface density over the past few decades, but consistent mean densities of ~350 kg m$^{-3}$ in the top 1 m have been observed around Dome Fuji in 1968, 1994, and 2007 CE (Endo*

*and Fujiwara, 1973; Shiraiwa et al., 1996; Sugiyama et al., 2012), implying that the variability of surface density around Dome Fuji has been small over the past half-century.*".

L414. Another aspect is the significance of such a low value. How to conclude based on a single core (~<10 cm in diameter) considering the randomness due to the wind ? The presence of long dunes as evidenced at Dome C (Picard et al. 2014)   may also jeopardize any interpretation of a few cores.

We agree with the comment that our statement was rather subjective on the interpretation of low values at ~1.5 m. We will remove the sentences in L410–413 and modify "*density fluctuations*" to "*lack of mean density increase*" in L414 (see reply to the previous comment).

L424. This is true up to ~ 50 m, as demonstrated in Fujita et al. (2009)

We will remove "*may*" in L424 and add a reference, Fujita et al. (2009).

L426. "Delta SSA" missing Delta in the parenthesis

We will correct it.

L454 "The intense IHDFs are formed by wind-packing (Koerner, 1971; Fujita et al., 2009), and their densities depend on wind speed (e.g., Sugiyama et al., 2012). Thus, wind is probably a key environmental factor controlling density variability (Fig. 6a)."   if this is true, some correlation in the density profiles between the nearby sites should be seen. This can be tested with this dataset. An alternative hypothesis is that the IHDF and ILDF are due to local randomness, i.e. due to the deposition of ten-centimeter thick patches of snow as shown in Picard et al. 2019. In such case, no correlation will be see in kilometer-distant cores.

We tested the correlation of the density profiles between the nearby sites (NDF18, NDF13, and NDFN) along the age scales (Figure below). There is no significant correlation among the three sites ($r = 0$–$0.15$), suggesting that the IHDF and ILDF are due to local randomness. We meant in the original text, as your latter hypothesis, that severe wind conditions (e.g., high wind speed and high wind gust frequency) enhance snow redistribution and the formation of IHDF, increasing horizontal (in turn, vertical) density variability. To make this point clear, we will revise the sentence as follows:

"*Thus, wind environment, such as mean wind speed or wind gust frequency, is probably a key factor controlling density variability (Fig. 6a) by affecting the frequency of snow redistribution and the degree of packing at the surface*."

[Figure]

Figure 9. Is is possible to add accumulation rate, either on the map or as value in the graphs.

We will add the accumulation rate as the value in the graph, for sites where the data is available for 1885–1992 CE: NDF, NDFN, DFS, DFSE, and DFNW as in Figure 1 and Table 1.

L528. an inverse correlation → negative correlation

We will correct it.

L 578-568. This part is less credible than the remaining of the analysis.

The visual determination of the maximums and minimums is subjective, especially the maximums in this particular case.

The years of maximum and minimum will be written as a range rather than a single value. We will also replace the phrase "*The six cores tend to show their maxima and minima in Δε around similar ages*" with "*Some maxima and minima of the Δε_{det} for the six cores appear at similar ages*"

The MC method is not detailed enough, especially how the temporal correlation in the series is taken into account (the 8-year moving average introduces very significant correlation).

We will extend the description of this method (see the modified sentences below). This analysis does not take into account the correlation coefficient because there is no statistical meaning in the average value of correlation coefficients for the 1000 data set, which includes non-significant correlation coefficients. Instead, we calculated the mean slope for the 1000 data set to determine the significance of the negative correlation. (We will add information on the correlation coefficient before this analysis is described; see below).

Also given that six cores are available, the method should start from this raw material, not from their average.

We will revise accordingly; we will first detrend the 0.5 m moving average of $\Delta\varepsilon$ for each core and investigate their correlation with the 8-year mean accumulation rate.

We will revise this part as follows.

[revised manuscript text omitted]

L612. "And the minor role of sintering for densification.". I don't understand.
This phrase was meant to point out that the sintering process, which is a major process to decrease SSA at deeper depths (> ~550 kg m$^{-1}$), is not important until 10 m. However, it was misplaced and confusing, thus we will remove it.

Data availability: the site is not accessible.

The data was indeed uploaded to the ADS website by the initial submission of our manuscript, but the ADS server was under maintenance from Oct. 5th to 13th, according to the email announcement from the management team. We are sorry that the maintenance notice was not displayed to the users who visited 440 the ADS site, but it should be accessible now.

https://ads.nipr.ac.jp/data/meta/A20230807-001

References:

[revised manuscript text omitted]